# Achieving Linear Convergence with Parameter-Free Algorithms in Decentralized Optimization

**Ilya Kuruzov**
Innopolis University
kuruzov.ia@phystech.edu.

**Gesualdo Scutari**
Purdue University
gscutari@purdue.edu.

**Alexander Gasnikov**
Innopolis University
gasnikov@yandex.ru

## Abstract

This paper addresses the minimization of the sum of strongly convex, smooth functions over a network of agents without a centralized server. Existing decentralized algorithms require knowledge of functions and network parameters, such as the Lipschitz constant of the global gradient and/or network connectivity, for hyperparameter tuning. Agents usually cannot access this information, leading to conservative selections and slow convergence or divergence. This paper introduces a decentralized algorithm that eliminates the need for specific parameter tuning. Our approach employs an operator splitting technique with a novel variable metric, enabling a local backtracking line-search to adaptively select the stepsize without global information or extensive communications. This results in favorable convergence guarantees and dependence on optimization and network parameters compared to existing nonadaptive methods. Notably, our method is the first *adaptive* decentralized algorithm that achieves linear convergence for strongly convex, smooth objectives. Preliminary numerical experiments support our theoretical findings, demonstrating superior performance in convergence speed and scalability.

## 1 Introduction

We study optimization across a network of $m > 1$ agents, modeled as an undirected, static graph, possibly with no centralized server. The agents cooperatively solve the following problem:

$$\min_{x \in \mathbb{R}^n} \sum_{i=1}^{m} f_i(x), \tag{P}$$

where $f_i : \mathbb{R}^n \to \mathbb{R}$ is the loss function of agent $i$, assumed to be strongly convex and smooth (i.e., with gradient being Lipschitz continuous), and accessible only to agent $i$.

This formulation applies to various fields, particularly emphasizing decentralized machine learning problems where datasets are produced and collected at different locations. Traditionally, statistical and computational methods in this domain have relied on a centralized paradigm, aggregating computational resources at a single, central location. However, this approach is increasingly unsuitable for modern applications with many machines, leading to server congestion, inefficient communication, and high energy consumption [27, 23]. This has motivated the surge of learning algorithms that target *decentralized* networks with *no servers*, a.k.a. *mesh* networks, which is the setting of this paper.

Decentralized convex optimization has a long history, with numerous algorithms applicable to Problem (P); recent tutorials include [34, 41, 6, 33, 45]. **Lack of adaptivity:** These methods share the hurdle of relying sensibly on the tuning of hyperparameters, such as the stepsize (a.k.a. learning rate), for both theoretical and practical convergence. Existing theories ensure convergence under generally conservative bounds on the stepsize, which depend on parameters like the Lipschitz constant of the global gradient, the spectral gap of the graph adjacency matrix, or other topological properties. Acquiring such information is challenging in practice, due to physical or privacy limitations and computational/communication constraints. This often leads to manual tuning, which is not only tedious but also results in less predictable, problem-dependent, and non-reproducible performance.

38th Conference on Neural Information Processing Systems (NeurIPS 2024).

**Parameter-free centralized methods:** On the the hand, significant progress has been made in the *centralized* setting to automate the selection of the stepsize across various optimization and learning problem classes. **(i)** Traditional approaches in optimization–such as line-search methods [36], Barzilai-Borwein's stepsize [3], and Polyak's stepsize [37]–have been supplemented by recent adaptive stepsize rules based on estimates of local curvature [30] and subsequent techniques [31, 19, 20, 22, 50]. **(ii)** In the ML community, adaptive gradient methods such as AdaGrad [11], Adam [18], AMSGrad [40], NSGD-M [9], and variants [25, 44, 29] have gained significant attention for training large-scale learning models. These methods apply to stochastic, nonconvex optimization problems. **(iii)** Further advancements extend adaptivity to stochastic/online convex optimization problems, e.g., [5, 15].

**Distributed adaptive methods:** While variant of these centralized algorithms have been adapted to federated architectures (server-client systems), e.g., in [39, 24, 8], their application to mesh networks is *not feasible*. In federated learning, a central server aggregates local model updates, a process integral to its hierarchical structure. However, mesh networks, which lack a centralized coordinating node, do not support such a direct aggregation of large-scale vectors. Recent attempts to implement some form of stepsize adaptivity for *stochastic (non)convex/online* optimization problems over mesh networks are [32, 7, 21]. These methods generally achieve adaptivity by properly normalizing agents' gradients using past information. However, with the exception of [21], they rely on the strong assumption that the (population) losses are *globally* Lipschitz continuous (i.e., their gradients are bounded). In fact, Lipschitz continuity in convex optimization readily unlocks parameter-free convergence by using stepsize tuning of $\mathcal{O}(1/\sqrt{k})$ (here, $k$ is the iteration index). Moreover, [32, 7] still require knowledge of some optimization parameters for the stepsize tuning, to guarantee convergence.

Attempts to introduce adaptivity in decentralized optimization for solving (P) have been explored in [12, 14, 13]. These methods bring the Barzilai-Borwein (BB)'s stepsize strategy into gradient tracking algorithms [43, 28, 35, 48]. **(i)** However, convergence of these algorithms *is not guaranteed* under the proposed BB strategy, unless the stepsizes *remain uniformly bounded from below and above* throughout the algorithm's trajectory–a condition the BB rule *does not inherently satisfy* in decentralized settings. Furthermore, these bounds for the stepsizes are typically unknown to the agents, as they depend on the strong convexity and smoothness constants of all agents' losses. Even with such knowledge, enforcing these conservative bounds contradicts the principle of adaptivity by potentially negating the advantages of a variable stepsize strategy that adapts based on local loss curvature, producing stepsize values significantly larger than theoretical thresholds used in nonadaptive methods. **(ii)** Additionally, to ensure contraction of the iterates, studies such as [13, 14] require multiple rounds of communications per iteration (gradient evaluation)–this demands the knowledge of network and optimization parameters at the agents' sides, making practical implementation unfeasible. **(iii)** None of these studies offer expressions of convergent rates for the explored algorithms, leaving it unclear whether the BB stepsize rule can provably outperform nonadaptive methods. **(iv)** Lastly, the methods discussed employ the traditional BB rule, which is only proven in centralized settings to produce convergence methods when minimizing *quadratic* losses. Simulations in [12, 13] are in fact performed only on quadratic functions.

**Open questions and challenges:** To our knowledge, no deterministic, parameter-free decentralized algorithms exist that solve Problem (P) over mesh networks, particularly achieving linear convergence when agents' functions are strongly convex and smooth. The current decentralized adaptive stochastic methods [32, 7, 21] discussed earlier do not adequately bridge this gap. Tailored for stochastic environments, these methods merely ensure that cumulative consensus errors along the iterations remain bounded, *not necessarily decreasing*. This typically involves either diminishing stepsizes or adjustments based on the final horizon to manage the bias-variance trade-off. These strategies fall short in deterministic scenarios like Problem (P), failing to ensure convergence to *exact* solutions, and achieve faster $\mathcal{O}(1/k)$ convergence rates in convex cases or *linear* rates in strongly convex scenarios.

**Major contributions:** This paper addresses this open problem. Our contributions are the following:

*1. A new parameter-free decentralized algorithm:* We propose a decentralized algorithm that eliminates the need for specific tuning of the stepsize. Our approach leverages a Forward-Backward operator splitting technique combined with a novel variable metric, enabling a local backtracking line-search procedure to adaptively select the stepsize at each iteration without requiring global information on optimization and network parameters or extensive communications. We are not aware of any other provable decentralized line-search methods over mesh networks.

Designing decentralized line-search procedures that are well-defined (terminating in a finite number of steps), locally implementable, and ensure algorithm convergence through satisfactory descent on an appropriate merit function presents significant challenges. A major issue is that line-search procedures merely based on the local curvature of agents' functions often fail to ensure convergence, producing *excessively large*, heterogeneous stepsizes that, e.g., poorly connected networks cannot support. This necessitates the identification of line-search *directions* and *surrogate functions* that encapsulate *both* optimization and network influences, aspects that have not yet formalized. Our design guidelines (cf., Sec. 3) are of independent interest; hopefully they will provide valuable insights for the development of other decentralized adaptive schemes, such as those based on alternative operator splittings.

*2. Convergence guarantees:* We have established linear convergence for the proposed decentralized adaptive method. Our analysis identifies critical quantities that capture the interplay between optimization conditions and network topology, directly influencing the convergence rates. Specifically: (a) In "well-connected" networks, the convergence rate exhibits a *separation property*: the overall rate is dictated by the slower of either the centralized gradient algorithm solving the same problem or a consensus algorithm run on the same mesh network. (b) Conversely, in "poorly" connected networks, the separation property vanishes, and the convergence rates are adversely affected by network degradation terms, still exhibiting a linear dependence on the condition number of the optimization loss. **(ii)** Unlike many existing distributed optimization frameworks, the optimization parameters in our rate expressions–such as smoothness and strong convexity constants–are localized to the *convex hull* of the traveled iterates. This localization arises from our adaptive stepsize strategy, which employs a line-search procedure tailored to local geometries, yielding more favorable dependencies on optimization parameters and thus enhanced convergence guarantees. **(iii)** Numerical experiments demonstrate superior performance of the proposed adaptive algorithm in convergence speed and scalability compared to existing non-adaptive methods.

## 1.1 Notation and paper organization

Capital letters denote matrices. Bold capital letters represent matrices where each row is an agent's variable, e.g., $\mathbf{X} = [x_1, \dots, x_m]^\top$. For such matrices, the $i$-th row is denoted by the corresponding lowercase letter with the subscript $i$; e.g., for $\mathbf{X}$, we write $x_i$ (as column vector). Let $\mathbb{S}^m$, $\mathbb{S}^m_+$, and $\mathbb{S}^m_{++}$ be the set of $m \times m$ (real) symmetric, symmetric positive semidefinite, and symmetric positive definite matrices, respectively; $A^\dagger$ denotes the Moore-Penrose pseudoinverse of $A$. The eigenvalues of $W \in \mathbb{S}^m$ are ordered in nonincreasing order, and denoted by $\lambda_1(W) \geq \cdots \geq \lambda_m(W)$. For two operators $A$ and $B$ of appropriate size, $(A \circ B)(\bullet)$ stands for $A(B(\bullet))$. We denote: $[m] = \{1, \dots, m\}$, for any integer $m \geq 1$; $[x]_+ := \max(x, 0)$, $x \in \mathbb{R}$; $1_m \in \mathbb{R}^m$ is the vector of all ones; $I_m$ (resp. $0_m$) is the $m \times m$ identity (resp. the $m \times m$ zero) matrix; the information on the dimension is omitted when not necessary; $\mathtt{null}(A)$ (resp. $\mathtt{span}(A)$) is the nullspace (resp. range space) of the matrix $A$. Let $\langle X, Y \rangle := \mathtt{tr}(X^\top Y)$, for any $X$ and $Y$ of suitable size ($\mathtt{tr}(\bullet)$) is the trace operator; and $\|X\|_M := \sqrt{\langle MX, X \rangle}$, for any symmetric, positive definite $M$ and $X$ of suitable dimensions. We still use $\|X\|_M$ when $M$ is positive semidefinite and $X \in \mathtt{span}(M)$.

# 2 Problem Setup

We investigate Problem (P) over a network of $[m]$ agents, modeled as an undirected, static, connected graph $\mathcal{G} = ([m], \mathcal{E})$, where $(i, j) \in \mathcal{E}$ if there is communication link (edge) between $i$ and $j$. For each agent $i$, we define by $\mathcal{N}_i := \{j : | (i, j) \in \mathcal{E}, \text{ for some } i \in [m]\} \cup \{i\}$ the set of immediate neighbors of agent $i$ (including agent $i$ itself).

**Assumption 1.** **(i)** *Each function $f_i$ in (P) is $L$-smooth and $\mu$-strong convex on $\mathbb{R}^n$, for some $L \in (0, \infty)$ and $\mu \in (0, \infty)$; and* **(ii)** *each agent $i$ has access only to its own function $f_i$.*

The following matrices are commonly utilized in the design of gossip-based algorithms.

**Definition 2** (Gossip matrices). *Let $\mathcal{W}_\mathcal{G}$ denote the set of matrices $\widetilde{W} = [\widetilde{W}_{ij}]_{i,j=1}^m$ that satisfy the following properties:* **(i)** *(compliance with $\mathcal{G}$) $\widetilde{W}_{ij} > 0$ if $(i, j) \in \mathcal{E}$; otherwise $\widetilde{W}_{ij} = 0$. Furthermore, $\widetilde{W}_{ii} > 0$, for all $i \in [m]$; and* **(ii)** *(doubly stochastic) $\widetilde{W} \in \mathbb{S}^m$ and $\widetilde{W}1_m = 1_m$.*

These matrices are standard in the literature on decentralized optimization algorithms, and several instances have been employed in practice; see [34, 41, 33] for some representative examples. Notice

that for any $\widetilde{W} \in \mathcal{W}_{\mathcal{G}}$ (assuming $\mathcal{G}$ connected) it hold: **(i)** (null space condition) $\texttt{null}(I_m - W) = \texttt{span}(1_m)$; and **(ii)** (eigen-spectrum distribution) $2I \succeq \widetilde{W} + I \succ 0_m$.

## 3   Algorithm Design

Our approach to solving Problem (P) involves a saddle-point reformulation tackled via a variable metric operator splitting, implementable across the graph $\mathcal{G}$. The innovative aspect of the proposed method lies in the selection of the variable metric that, coupled with a Forward Backward Splitting (FBS), enable adaptive stepsize selections through a decentralized line-search procedures.

Introducing local copies $x_i \in \mathbb{R}^d$ of the shared variable $x$ (the $i$-th one is controlled by agent $i$), and the stack matrix $\mathbf{X} := [x_1, \dots, x_m]^\top \in \mathbb{R}^{m \times n}$, let us consider the following auxiliary problem:

$$\min_{\mathbf{X} \in \mathbb{R}^{m \times n}} \left[ F(K\mathbf{X}) := \sum_{i=1}^m f_i([K\mathbf{X}]_i) \right], \quad \text{s.t. } \text{Ł} \, \mathbf{X} = 0. \tag{P$'$}$$

Here, Ł and $K$ are $m \times m$ matrices that meet the following criteria: **(c1)** $\text{Ł} \in \mathbb{S}^m$ and $\texttt{null}(\text{Ł}) = \texttt{span}(1_m)$; **(c2)** $K \in \mathbb{S}_{++}^m$ and $\texttt{null}(I - K) = \texttt{span}(1_m)$; and **(c3)** Ł and $K$ commute. Conditions (c1) and (c2) ensure that (P) and (P$'$) are equivalent. Specifically, any solution $\mathbf{X}^\star$ of (P$'$) has the form of $\mathbf{X}^\star = 1_m(x^\star)^\top$, where $x^\star$ solves (P), and vice versa. While not essential, condition (c3) is postulated to simplify the algorithm derivation.

Primal-dual optimality for (P$'$) reads, with $\mathbf{Y}$ being the dual-variable associated with the constraints,

$$(A + B) \left( \begin{bmatrix} \mathbf{X}^\star \\ \mathbf{Y}^\star \end{bmatrix} \right) = 0, \quad \text{where} \quad A := \begin{bmatrix} K \circ \nabla F \circ K & 0 \\ 0 & 0 \end{bmatrix} \text{ and } B := \begin{bmatrix} 0 & \text{Ł} \\ -\text{Ł} & 0 \end{bmatrix}.$$

Given $\mathbf{X}^k, \mathbf{Y}^k$ at iteration $k$, the update $\mathbf{X}^{k+1}, \mathbf{Y}^{k+1}$ via FBS with metric $C \in \mathbb{S}_{++}^{2m}$ reads [4]

$$(C + B) \left( \begin{bmatrix} \mathbf{X}^{k+1} \\ \mathbf{Y}^{k+1} \end{bmatrix} \right) = (C - A) \left( \begin{bmatrix} \mathbf{X}^k \\ \mathbf{Y}^k \end{bmatrix} \right). \tag{1}$$

Monotone operator theory [4] ensures convergence of (1) under the following conditions:

**(c4)** $B$ is a monotone operator, $C \in \mathbb{S}_{++}^{2m}$, and **(c5)** $I - C^{-1/2} A C^{-1/2}$ is an averaged operator.

Condition (c4) is satisfied by construction; (c5) can be enforced through a suitable selection of $C \in \mathbb{S}_{++}^{2m}$ while leveraging the co-coercivity of $A$ (implied by Assumption 1). Denoting by $\alpha > 0$ the stepsize employed in the algorithm, we seek for $C$ with the following structure:

$$C = \begin{bmatrix} \alpha^{-1} C_1 & 0 \\ 0 & C_2 \end{bmatrix}, \quad \text{with} \quad C_1, C_2 \in \mathbb{S}_{++}^m$$

to be determined. We proceed solving (1). Taking $(C + B)^{-1}$, we have

$$\begin{aligned}
\mathbf{X}^{k+1} &= (I) (\mathbf{X}^k) - \alpha \left( (II) (\mathbf{X}^k) + (III) (\mathbf{Y}^k) \right), \\
\mathbf{Y}^{k+1} &= (IV) (\mathbf{Y}^k) + (V) (\mathbf{X}^k),
\end{aligned} \tag{2}$$

where

$$\begin{aligned}
(I) &:= I_m - \alpha \cdot C_1^{-1} \text{Ł} \left( C_2 + \alpha \cdot \text{Ł} \, C_1^{-1} \text{Ł} \right)^{-1} \text{Ł}, \\
(II) &:= (I) \, C_1^{-1} K \, \nabla F \circ K, \\
(III) &:= C_1^{-1} \text{Ł} \left( C_2 + \alpha \cdot \text{Ł} \, C_1^{-1} \text{Ł} \right)^{-1} C_2, \\
(IV) &:= \left( C_2 + \alpha \cdot \text{Ł} \, C_1^{-1} \text{Ł} \right)^{-1} C_2, \\
(V) &:= \left( C_2 + \alpha \cdot \text{Ł} \, C_1^{-1} \text{Ł} \right)^{-1} \text{Ł} \left( I - \alpha \cdot C_1^{-1} K \, \nabla F \circ K \right).
\end{aligned} \tag{3}$$

In addition to satisfying (c5), $C_1, C_2 \in \mathbb{S}_{++}^m$ must be strategically chosen to facilitate the design of a decentralized line-search procedure for $\alpha$. We propose the following guiding principles:

**(c6)** The range of admissible stepsize values $\alpha$ ensuring convergence–hence satisfying (c5)–should be independent of the network parameters; and

**(c7)** the operators $(I)$, $(II)$, and $(III)$ in (2) should be independent of $\alpha$.

At a high level, (c6) aims to decouple the line-search mechanism from network-dependent constraints. By doing so, it ensures that performing the line-search from the agents' sides requires no mid-process communications during backtracking, relying solely on local computations. Meanwhile, (c7) facilitates the identification of $-((II)(\mathbf{X}^k) + (III)(\mathbf{Y}^k))$ in (2) as a potential line-search direction. This direction must be paired with an appropriate surrogate function, which we will define shortly.

Among several potential selections, in this paper, we consider the following for $C_1$ and $C_2$:

$$C_1 = K \quad \text{and} \quad C_2 = \alpha K^{-1}\left(c^{-1} I - Ł^2\right), \text{ with } c < 1/2, \tag{4}$$

which satisfy all the specified requirements. Using (4) and (c3), the operators in (3) simplify to

$$(I) = I_m - c{\cdot}Ł^2, \ \ (II) = (I)\nabla F \circ K, \ \ (III) = Ł K^{-1}, \ \ (IV) = (I), \ \ (V) = \frac{c}{\alpha}{\cdot}K \, Ł \, (I - \alpha \nabla F \circ K).$$

Notice that $(I), (II)$, and $(III)$ are independent of the stepsize. Substituting the above expressions in (2) and introducing $\mathbf{D}^k := K^{-1}Ł\mathbf{Y}^k$, the algorithm can be rewritten as

$$\mathbf{X}^{k+1} = (I - cŁ^2)\,\mathbf{X}^k - \alpha \cdot (I - cŁ^2)\left(\mathbf{D}^k + \nabla F(K\mathbf{X}^k)\right),$$

$$\mathbf{D}^{k+1} = (I - cŁ^2)\,\mathbf{D}^k + \frac{c}{\alpha} \cdot Ł^2\left(\mathbf{X}^k - \alpha \nabla F(K\mathbf{X}^k)\right).$$

To make the above updates compliant with the graph $\mathcal{G}$ while satisfying (c1)-(c3), we set $Ł^2 = (I - \widetilde{W})$, with $\widetilde{W} \in \mathcal{W}_{\mathcal{G}}$, and $K = I - cŁ^2$, where $c \in (0, 1/2)$ is a free universal constant. Introducing $W := (1-c)I_m + c\widetilde{W} \in \mathcal{W}_{\mathcal{G}}$, the final decentralized algorithm can be rewritten as

$$\mathbf{X}^{k+1/2} = W\,\mathbf{X}^k, \quad \mathbf{D}^{k+1/2} = W\left(\mathbf{D}^k + \nabla F(\mathbf{X}^{k+1/2})\right),$$

$$\mathbf{X}^{k+1} = \mathbf{X}^{k+1/2} - \alpha \cdot \mathbf{D}^{k+1/2}, \tag{5}$$

$$\mathbf{D}^{k+1} = \mathbf{D}^{k+1/2} + \frac{1}{\alpha} \cdot \left(\mathbf{X}^k - \mathbf{X}^{k+1/2} - \alpha \nabla F(\mathbf{X}^{k+1/2})\right).$$

Finally, it can be verified that (c6) is met if $(\sqrt{\alpha}K^{-1/2}) \circ \nabla F \circ (\sqrt{\alpha}K^{-1/2})$ is nonexpansive, which holds if $\alpha < 1/L$, being independent on the network parameters. Next, we introduce a line-search procedure that enables the use of an adaptive stepsize $\alpha$ rather than a constant one satisfying the above more conservative bound.

**Decentralized backtracking:** It is not difficult to check that $-\mathbf{D}^{k+1/2}$ is a descent direction of $F^k(\mathbf{X}) := F(\mathbf{X}) + \langle \mathbf{D}^k, \mathbf{X}\rangle$ at $\mathbf{X}^{k+1/2}$. This naturally suggests the following backtracking procedure for $\alpha$: at iteration $k$, find the largest $\alpha^k > 0$ such that

$$F^k(\mathbf{X}^{k+1}) \leq F^k(\mathbf{X}^{k+1/2}) + \left\langle \nabla F^k(\mathbf{X}^{k+1/2}), \mathbf{X}^{k+1} - \mathbf{X}^{k+1/2}\right\rangle + \frac{\delta}{2\alpha^k}\|\mathbf{X}^{k+1} - \mathbf{X}^{k+1/2}\|^2, \tag{6}$$

where $\delta \in (0, 1]$ is a tuning parameter. However, this condition would require a communication round for each backtracking step. To reduce the communication burden, we introduce a local stepsize for each agent $i$, denoted by $\alpha_i^k$, determined by a backtracking line-search on the local function $f_i^k(x) := f_i(x) + \langle d_i^k, x\rangle$. Specifically, each $\alpha_i^k$ is the largest positive value satisfying

$$f_i^k(x_i^{k+1}) \leq f_i^k(x_i^{k+1/2}) + \left\langle \nabla f_i^k(x_i^{k+1/2}), x_i^{k+1} - x_i^{k+1/2}\right\rangle + \frac{\delta}{2\alpha_i^k}\|x_i^{k+1} - x_i^{k+1/2}\|^2. \tag{7}$$

Clearly $\alpha^k = \min_{i \in [m]} \alpha_i^k$ also satisfies (6). Noticing that $f_i^k$ has the same smooth (and strong convexity) constant(s) of $f_i$, one can replace $f_i^k$ in (7) with $f_i$. The proposed decentralized algorithm is summarized in Algorithm 1, with the backtracking line-search procedure detailed in Algorithm 2.

## 3.1 Discussion

Several comments are in order.

**On the proposed algorithm:** We emphasize that selecting $K \neq I_m$ in (P$'$) marks a significant departure from the commonly used saddle-point reformulations of Problem (P), where $K = I_m$, e.g., [46, 34, 33, 1]. Choosing $K \neq I_m$, in conjunction with the novel variable metric $C$ in the FBS as

---

**Algorithm 1**

---

**Data:** (i) Initialization $\mathbf{X}^0 \in \mathbb{R}^{m \times n}$ and $\mathbf{D}^0 = 0$; (ii) initial value $\alpha_{-1} \in (0, \infty)$; (iii) Backtracking parameters $\delta \in (0, 1]0$; (iv) nondecreasing sequence $\{\gamma^k\}_k \subseteq [1, \infty)$ (v) Gossip matrix $W := (1-c)I_m + c\widetilde{W}$, with $\widetilde{W} \in \mathcal{W}_{\mathcal{G}}$, and $c \in (0, 1/2]$. Set the iteration index $k = 0$.

1: (S.1) `Communication step`: Agents updates primal and dual variables via gossiping:

$$\mathbf{X}^{k+1/2} = W\,\mathbf{X}^k \quad \text{and} \quad \mathbf{D}^{k+1/2} = W\left(\mathbf{D}^k + \nabla F(\mathbf{X}^{k+1/2})\right);$$

2: (S.2) `Decentralized line-search`: Each agent updates $\alpha_i^k$ according to

$$\alpha_i^k = \texttt{Backtracking}\left(\alpha^{k-1}, f_i, x_i^{k+1/2}, -d_i^{k+1/2}, \gamma^k, \delta\right);$$

3: (S.3) `Global min-consensus`:
$$\alpha^k = \min_{i \in [m]} \alpha_i^k;$$

4: (S.4) `Local updates of the primal and dual variables`:

$$\mathbf{X}^{k+1} = \mathbf{X}^{k+1/2} - \alpha^k \cdot \mathbf{D}^{k+1/2},$$

$$\mathbf{D}^{k+1} = \mathbf{D}^{k+1/2} + \frac{1}{\alpha^k} \cdot \left(\mathbf{X}^k - \mathbf{X}^{k+1/2} - \alpha^k \nabla F(\mathbf{X}^{k+1/2})\right).$$

5: (S.5) If a termination criterion is not met, $k \leftarrow k+1$ and go to step (S.1).

---

**Algorithm 2** Backtracking($\alpha, f, x, d, \gamma, \delta$)

---

1: $\alpha^+ := \gamma\alpha$;
2: $x^+ := x + \alpha^+ d$; set $t = 1$;
3: **while** $f(x^+) > f(x) + \langle \nabla f(x), x^+ - x \rangle + \frac{\delta}{2\alpha^+}\|x^+ - x\|^2$ **do**
4:    $\alpha^+ \leftarrow (1/2)\alpha^+$;
5:    $x^+ := x + \alpha^+ d$;
6:    $t \leftarrow t + 1$;
   **return** $\alpha^+$.

---

specified in (4), is critical to obtain a valid line-search procedure that is also implementable across the network. For instance, popular decentralized algorithms such as EXTRA [42] and NIDS [26] can be interpreted as FBS with suitable metrics associated with the primal-dual reformulation of (P) as (P′) but with $K = I_m$. However, these schemes do not facilitate any suitable line-search, as no stepsize-independent descent direction can be identified in their updates. Hopefully, our approach will provide principled guidelines for the design of other parameter-free decentralized algorithms, stemming from alternative decentralized formulations of (P) and their corresponding operator splittings.

**On the backtracking:** The following Lemma shows that the line-search procedure in Algorithm 2 is well-defined, as long as the function $f$ therein is locally smooth.

**Lemma 3.** *Let $f$ in Algorithm 2 be any $L_f$-smooth and $\mu_f$-strongly convex function on the segment $[x, x + \gamma\alpha d]$, where $L_f \in (0, \infty)$, $\mu_f \in [0, \infty)$, and $\gamma \in [1, \infty)$. The following hold for Algorithm 2:*
*(i) The returned $\alpha^+$ satisfies*

$$\min\left(\gamma\alpha, \frac{\delta}{2L_f}\right) \leq \alpha^+ \leq \min\left(\gamma\alpha, \frac{\delta}{\mu_f}\right) \leq \infty. \tag{8}$$

*Therefore, the backtracking procedure terminates in at most $\max\left(1, \lceil \log_2 \frac{2L_i\gamma\alpha}{\delta}\rceil\right)$ t-steps;*

*(ii) For any $\alpha^+$ returned by Algorithm 2, any $\bar{\alpha}^+ \in (0, \alpha^+]$ also satisfies the backtracking condition.*

Notice that the last statement of the lemma guarantees that the each $\alpha^k = \min_{i \in [m]} \alpha_i^k$ satisfies the descent property (6) on the global loss $F^k$, as each $\alpha_i^k$ meets the local condition (7).

The sequence $\{\gamma^k\}_{k=1}^{\infty}$ used in line 1 of the backtracking algorithm, with each $\gamma^k \geq 1$, is introduced to favor nonmonotone, and thus potentially larger, stepsize values between two consecutive line-

search calls. Any sequence satisfying $\gamma^k \downarrow 1$ and $\prod_{k=1}^{\infty} \gamma^k = \infty$, is advisable. In our experiments, we found the following rule quite effective: $\gamma^k = \left((k+\beta_1)/(k+1)\right)^{\beta_2}$, for some $\beta_2 > 0$ and $\beta_1 \geq 1$. One can also opt for $\gamma^k = 1$, for all $k$, thus eliminating this extra parameter, if simplicity is desired.

**On the min-consensus:** Step (S.3) involves a min-consensus across the network to establish a common stepsize, $\alpha^k = \min_{i \in [m]} \alpha_i^k$, among the agents. This procedure is easily implemented in federated systems, where a server node facilitates information exchange between clients. Interestingly, this min-consensus protocol is also well-suited to current wireless mesh network technologies. Modern networks support multi-interface communications, including WiFi and LoRa (Low-Range) [17, 2, 16]. WiFi allows high-speed, short-range communications, supporting a mesh topology where nodes transmit large data volumes to immediate neighbors. Conversely, LoRa facilitates long-range but low-rate communications, ideal for communication flooding that reaches all network nodes in a single hop but transmits minimal information. Therefore, in multi-interface networks, the proposed algorithm operates by transmitting vector variables in Steps (S.1) via WiFi, while LoRa is used for the min-consensus in Step (S.3). Furthermore, the values $\alpha_i^k$'s can be quantized to their nearest lower values using a few bits before transmission. Based on Lemma 3(ii), this quantization ensures that the descent condition (6) is still met with the resultant min quantized stepsize. This approach renders the extra communication cost for implementing the global min-consensus step negligible.

For networks where LoRa technology cannot be used, Sec. 5 proposes a variant of Algorithm 1 wherein the global min-consensus step (S.3) is replaced by a local min-consensus procedure.

## 4    Convergence Results

We begin introducing a quantity of interest that helps identifying different operational regimes of the proposed algorithm. Let $(\mathbf{X}^\star, \mathbf{D}^\star)$ be a fixed point of Algorithm 1 (whose existence is ensured by Assumption 1), and let $\{(\mathbf{X}^k, \mathbf{D}^k)\}$ be the iterates generated by Algorithm 1. Define

$$r^k = \frac{\sqrt{\frac{1}{(\alpha^k)^2}\|\mathbf{X}^k\|^2_{c(I-\widetilde{W})} + \left\|c(I-\widetilde{W})\left(\nabla F(\mathbf{X}^{k+1/2}) + \mathbf{D}^k\right)\right\|^2_M}}{\max\left(\frac{1}{\alpha^k}\|\mathbf{X}^k - \mathbf{X}^\star\|, \|\mathbf{D}^k - \mathbf{D}^\star\|_M\right)}, \text{ with } M := c^{-1}(I-\widetilde{W})^\dagger - I. \tag{9}$$

The following comments are in order. **(i)** Both $\mathbf{D}^k$ and $\mathbf{D}^\star$ lie in the $\operatorname{span}(I - \widetilde{W}) = \operatorname{span}(I - W)$, for all $k$, and $M$ is positive defined on this span. Consequently, $\|\mathbf{D}^k - \mathbf{D}^\star\|_M > 0$ for all $\mathbf{D}^k \neq \mathbf{D}^\star$, and $\|\mathbf{D}^k - \mathbf{D}^\star\|_M = 0$ if and only if $\mathbf{D}^k = \mathbf{D}^\star$. **(ii)** Under Assumption 1, $\mathbf{X}^\star = 1(x^\star)^\top$, where $x^\star$ is the solution of (P). **(iii)** The quantity $r^k$ reflects the algorithm's convergence progress through the evolution of the dual variables and consensus error. Rewriting the update of the dual variables as $\mathbf{D}^{k+1} = \mathbf{D}^k + \frac{c}{\alpha^k}(I - \widetilde{W})\mathbf{X}^k - c(I - \widetilde{W})\left(\nabla F(\mathbf{X}^{k+1/2}) + \mathbf{D}^k\right)$, we observe that small values of $\|\frac{c}{\alpha^k}(I-\widetilde{W})\mathbf{X}^k - c(I-\widetilde{W})\left(\nabla F(\mathbf{X}^{k+1/2}) + \mathbf{D}^k\right)\|$ compared to $\|\mathbf{D}^k - \mathbf{D}^\star\|$ and $\|\mathbf{X}^k - \mathbf{X}^\star\|$–hence small $r^k$ values–indicate slow convergence improvements (see Lemma 8 in the appendix).

We remark that $r^k$ need not be known by the agents; it is instrumental only for the analysis and posterior assessment of algorithm performance.

Linear convergence is established below via contraction of the following merit function

$$V^k := \left\|\mathbf{X}^k - \mathbf{X}^\star\right\|^2 + (\alpha^{k-1})^2 \|\mathbf{D}^k - \mathbf{D}^\star\|^2_M. \tag{10}$$

**Theorem 4.** *Given Problem* (P) *under Assumption 1, let* $\{(\mathbf{X}^k, \mathbf{D}^k)\}$ *be the iterates generated by Algorithm 1. Then, the following holds*

$$V^{k+1} \leq \max\left(\frac{\alpha^k}{\alpha^{k-1}}, 1\right)^2 \left(1 - \min\left(\rho_1^k, \delta(r^k)^2\right)\right) V^k, \text{ where } \rho_1^k := \frac{\mu^k \alpha^k}{2}(1 - c(1 - \lambda_m(\widetilde{W})))^2 < 1. \tag{11}$$

*If* $r^k < \sqrt{2}/4$, *then*

$$V^{k+2} \leq \max\left(\frac{\alpha^{k+1}}{\alpha^k}, 1\right)^2 \max\left(\frac{\alpha^k}{\alpha^{k-1}}, 1\right)^2 \left(1 - \rho_2^k\right) V^k, \tag{12}$$

*where*

$$\rho_2^k := \frac{(1 - c(1 - \lambda_m(\widetilde{W})))^2}{128(\gamma^k)^2 \max(1, \lambda_{\max}(M))} \min\left(\mu^{k+1}\alpha^{k+1}, \mu^k\alpha^k, \frac{1}{L^k\alpha^k}\right) < 1.$$

*Here $\mu^k$ (resp. $\mu^{k+1}$) and $L^k$ are the strong convexity and smoothness constants of (each) $f_i$ along the segment $[x_i^{k+1/2}, x^\star]$ (resp. $[x_i^{k+1+1/2}, x^\star]$), respectively.*

The theorem establishes linear convergence of Algorithm 1. As $\max(1, (\alpha^k/\alpha^{k-1})^2)$ is bounded away from zero and uniformly upper bounded (with value depending on the sequence $\{\gamma^k\}$)–see Lemma 3–the convergence rate is predominantly determined by $\{\rho_1^k\}$, $\{\rho_2^k\}$, and $\{r^k\}$. Notice that, in the setting of the theorem, each $\rho_1^k, \rho_2^k \in [0,1)$. Intriguingly, the algorithm exhibits different operational regimes based on the range of values $r^k$ takes along the traveled trajectory. At the high level, if $r^k$ remains "large", faster convergence can be guaranteed, as certified by (11); otherwise $V^k$ decreases every two consecutive iterations (see (12)), yielding to a slower convergence. The number of iterations required to reach a desired termination accuracy is given next.

**Corollary 4.1.** *Instate the setting of Theorem 4, with now $\{\gamma^k\}$ being chosen such that $\gamma^k \leq \left((k+\beta_1)/(k+1)\right)^{\beta_2}$, for all $k$ and some $\beta_1 \geq 1, \beta_2 > 0$. Then*

$$\left\|\mathbf{X}^{k+1} - \mathbf{X}^\star\right\|^2 + \frac{1}{4L^2}\|\mathbf{D}^{k+1} - \mathbf{D}^\star\|_M^2 \leq \varepsilon,$$

*for all $k \geq N_\varepsilon$, where $N_\varepsilon$ is given as follows:*

*If $r^k \geq r_{low} := \frac{1}{\sqrt{2}} \min\left(\frac{1}{2}, \frac{1}{\sqrt{\lambda_{\max}(M)}}\right)$, for all $k$,*

$$N_\varepsilon = \mathcal{O}\left(\frac{1}{\delta} \max\left(\frac{1}{c(1-\lambda_2(\widetilde{W}))}, \frac{\kappa}{(1-c(1-\lambda_m(\widetilde{W})))^2}\right) \log(V^0/\varepsilon)\right); \qquad (13)$$

*otherwise,*

$$N_\varepsilon = \mathcal{O}\left(\frac{1}{\delta} \frac{\kappa}{(1-c(1-\lambda_m(\widetilde{W})))^2 c(1-\lambda_2(\widetilde{W}))} \log(V^0/\varepsilon)\right). \qquad (14)$$

*Here $\kappa$ is the condition number of each $f_i$ restricted to the convex hull of $\{x^\star, \{x_i^k, x_i^{k+1/2}\}_{k=0}^{N_\varepsilon}\}$, and $\mathcal{O}$ hides the dependence on $\beta_1$ and $\beta_2$.*

Corollary 4.1 identifies the following two different operational regimes of the algorithm, resulting in difference performance based upon the network connectivity and optimization condition number.

**(1) Strong connectivity regime:** when $r^k \geq r_{low}$, for all $k$, a fact that numerically has been observed for 'relatively good' network connectivity, the convergence rate exhibits a separation in the dependence on the network and optimization parameters. Since $1 - c(1-\lambda_m(\widetilde{W})) > 1 - 2c$, it follows that, when $c(1-\lambda_2(\widetilde{W})) \geq (1-2c)/\sqrt{\kappa}$, $N_\varepsilon$ reduces to $\mathcal{O}(\kappa)$ (omitting the dependence on $\varepsilon$), which matches the complexity of the centralized gradient algorithm. This suggests scenarios where the optimization problem is harder than a consensus problem over the same network, resulting in the bottleneck between the two. Conversely, when the condition number $\kappa$ is large relative to the network connectivity $1 - \lambda_2(\widetilde{W})$, the rate is determined by that of the consensus algorithm running on the same network, that is, $\mathcal{O}((1-\lambda_2(\widetilde{W}))^{-1})$. The above rate separation property mirrors that of certain *nonadaptive* primal-dual decentralized schemes including NEXT [10], AugDGM [47], Exact Diffusion [49] (with rate expression as improved in [46]), NIDS [26], and ABC [46].

**(2) Worst-case regime:** This regime reflects the algorithm's worst-case performance, typically registered in "weakly" connected networks: the convergence rate reads $\mathcal{O}(\kappa/(1-\lambda_2(\widetilde{W})))$, where optimization and network parameters are now mixed. This rate aligns with those of *nonadaptive* decentralized gradient-tracking schemes, such as DGing [35], SONATA [43] (subject to sufficiently small network connectivity), and [38].

In summary, the convergence rate of Algorithm 1 resembles in the form that of existing nonadaptive decentralized methods, but offers more favorable dependence on the condition number than that typically found in those algorithms. Specifically, the condition number in (13) and (14) is the *local* condition number, defined on the convex hull of the trajectory, which is generally much smaller than the *global* condition number governing decentralized algorithms in the literature. This demonstrates the algorithm's capability to adapt to the local geometry of the optimization problem.

# 5 From Global to Local Min-Consensus

This section extends Algorithm 1 to settings where the global min-consensus procedure in (S.3) is not implementable. For these cases, we propose to replace such a step with a *local* min-consensus procedure. The new algorithm is formally described in Algorithm 3 and briefly commented next.

In step (S.3), each agent now computes its stepsize taking the minimum values among those of their immediate neighbors only (including itself). This produces possibly different stepsizes $\alpha_i^k$ for each agent, collected in the diagonal matrix $\Lambda^k = \mathtt{diag}(\alpha_1^k \dots \alpha_m^k)$. Because of that, in order to still guarantee $\mathbf{D}^k \in \mathtt{span}(I - \widetilde{W})$–a key property for the convergence of the algorithm–we slightly modified the updates of the dual variable in (S.4), compared with the same step in Algorithm 1. Specifically, the updating direction of the dual variable as in Algorithm 1, $(\alpha^k)^{-1}(\mathbf{X}^k - \mathbf{X}^{k+1/2} - \alpha^k \nabla F(\mathbf{X}^{k+1/2}))$, is replaced in Algorithm 3 by $(\Lambda^k)^{-1}\mathbf{X}^k - \mathbf{X}_\Lambda^{k+1/2} - \nabla F(\mathbf{X}^{k+1/2})$, where $\mathbf{X}_\Lambda^{k+1/2} = W(\Lambda^k)^{-1}\mathbf{X}^k$. Notice that, if all the stepsizes are equal, the update (S.4) in Algorithm 3 reduced to that in Algorithm 1. Finally, we point out that the computation of $\mathbf{X}_\Lambda^{k+1/2}$ requires only the extra communication of neighboring stepsizes (thus scalar) values, which has a negligible cost.

---

**Algorithm 3**

---

**Data:** (i) Initialization $\mathbf{X}^0 \in \mathbb{R}^{m \times n}$ and $\mathbf{D}^0 = 0$; (ii) initial value $\alpha_{-1} \in (0, \infty)$; (iii) Backtracking parameters $\delta \in (0, 1]$; (iv) nondecreasing sequence $\{\gamma^k\}_k \subseteq [1, \infty)$ (v) Gossip matrix $W := (1 - c)I_m + c\widetilde{W}$, with $\widetilde{W} \in \mathcal{W}_{\mathcal{G}}$, and $c \in (0, 1/2]$. Set the iteration index $k = 0$.

1: (S.1) Communication step: Agents updates primal and dual variables via gossiping:

$$\mathbf{X}^{k+1/2} = W\mathbf{X}^k \quad \text{and} \quad \mathbf{D}^{k+1/2} = W\left(\mathbf{D}^k + \nabla F(\mathbf{X}^{k+1/2})\right);$$

2: (S.2) Decentralized line-search: Each agent updates $\overline{\alpha}_i^k$ according to

$$\overline{\alpha}_i^k = \mathtt{Backtracking}\left(\alpha^{k-1}, f_i, x_i^{k+1/2}, d_i^{k+1/2}, \gamma^k, \delta\right);$$

3: (S.3) Local min-consensus:
$$\alpha_i^k = \min_{j \in \mathcal{N}_i} \overline{\alpha}_j^k, \quad \forall i \in [m];$$

    Define $\Lambda^k = \mathtt{diag}(\alpha_1^k \dots \alpha_m^k)$;

4: (S.4) Local updates of the primal and dual variables:

$$\mathbf{X}^{k+1} = \mathbf{X}^{k+1/2} - \Lambda^k \cdot \mathbf{D}^{k+1/2}, \quad \mathbf{X}_\Lambda^{k+1/2} = W(\Lambda^k)^{-1}\mathbf{X}^k,$$

$$\mathbf{D}^{k+1} = \mathbf{D}^{k+1/2} + (\Lambda^k)^{-1}\mathbf{X}^k - \mathbf{X}_\Lambda^{k+1/2} - \nabla F(\mathbf{X}^{k+1/2}).$$

5: (S.5) If a termination criterion is not met, $k \leftarrow k + 1$ and go to step (S.1).

---

Convergence of Algorithm 3 is established in the following theorem.

**Theorem 5.** *Instate assumptions in Theorem 4, applied now to Algorithm 3, with $\{\gamma^k\}$ being chosen such that $\gamma^k \leq \left((k + \beta_1)/(k + 1)\right)^{\beta_2}$, for all $k$ and some $\beta_1 \geq 1, \beta_2 > 0$. Further, suppose there exists a constant $R > 0$ such that $V^k \leq R$, for all $k$. Then*

$$\min_{j \in [1, N+1]} \left\|\mathbf{X}^j - \mathbf{X}^\star\right\|^2 + \frac{1}{4L^2}\|\mathbf{D}^j - \mathbf{D}^\star\|_M^2 \leq \varepsilon,$$

*with*

$$N = \mathcal{O}\left(\max\left(\log d_{\mathcal{G}} + \log N_\varepsilon, \log \alpha_0 L\right) \max(N_\varepsilon, d_{\mathcal{G}})\right),$$

*where $N_\varepsilon$ is defined as in Corollary 4.1 (replacing therein $V_0$ with $R$).*

Interestingly, Theorem 5 states that the degradation of the convergence rate when a local-min consensus is used instead of the global one is mild. Specifically, up to log factors, the total number of iterations to $\varepsilon$-optimality depends on $d_{\mathcal{G}}$ (the diameter of the graph $\mathcal{G}$), if $d_{\mathcal{G}} > N_\varepsilon$. This result is somehow expected, as min-consensus requires a number of iterations proportional to $d_{\mathcal{G}}$ to propagate through the entire network. However, monotonicity in the decrease of the primal and dual errors can no longer be guaranteed when min-consensus is employed.

# 6 Numerical Results

This section presents some preliminary numerical results. We compare Algorithm 1 and Algorithm 3 with EXTRA [42] and NIDS [26] on a ridge regression problem using synthetic data. Further experiments are presented in the appendix. All experiments are run on Acer Swift 5 SF514-55TA-56B6, with processor Intel(R) Core(TM) i5-8250U @ CPU 1.60GHz, 1800 MHz.

**Ridge regression:** It is an instance of (P), with $f_i(x) = \|A_i x_i - b_i\|^2 + \sigma \|x_i\|_2^2$, where we set $A_i \in \mathbb{R}^{20 \times 300}, b_i \in \mathbb{R}^{20}$, and $\sigma = 0.1$. The elements of $A_i, b_i$ are independently sampled from the standard normal distribution; the regularization is set to $\sigma = 0.1$. We simulated a network of $m = 20$ agents, and the following three different graph topologies, reflecting varying connectivity levels: **(i)** $\mathcal{G}_1$: Graph-path with $m - 1$ edges and diameter $m - 1$, i.e., $\mathcal{G} = \{[m], \{(i, i+1)\}_{i=1}^{m-1}\}$; **(ii)** $\mathcal{G}_2$: Erdős–Rényi graph, sparsely connected; and **(iii)** $\mathcal{G}_3$: Erdős–Rényi graph, well-connected.

Results are summarized in Fig. 1 and Fig. 2. For EXTRA and NIDS we use a grid-search tuning, chosen to achieve the best practical performance. Algorithm 1 and Algorithm 3 are simulated under the following choice of the line-search parameters satisfying Corollary 4.1: $\gamma^k = (k+2)/(k+1)$, $\delta = 1$. For all the algorithms we used the Metropolis-Hastings weight matrix $W \in \mathcal{G}_\mathcal{W}$ [34].

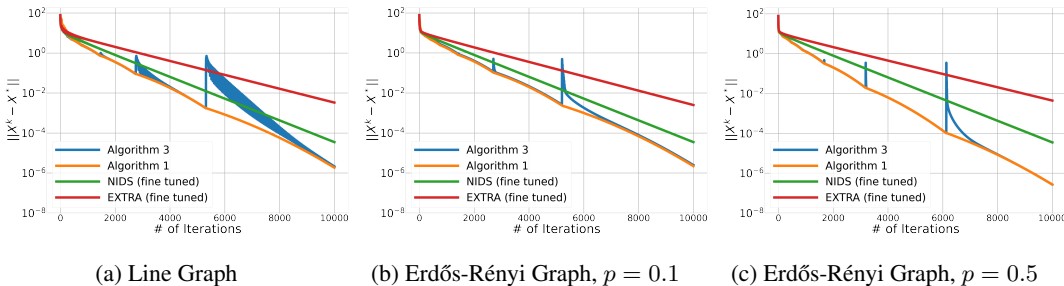

| (a) Line Graph | (b) Erdős-Rényi Graph, $p = 0.1$ | (c) Erdős-Rényi Graph, $p = 0.5$ |

Figure 1: **Ridge regression** on different graphs: (1a) Line graph; (1b) Erdős-Rényi Graph with edge activation probability $p = 0.1$; (1c) Erdős-Rényi Graph with edge activation probability $p = 0.5$

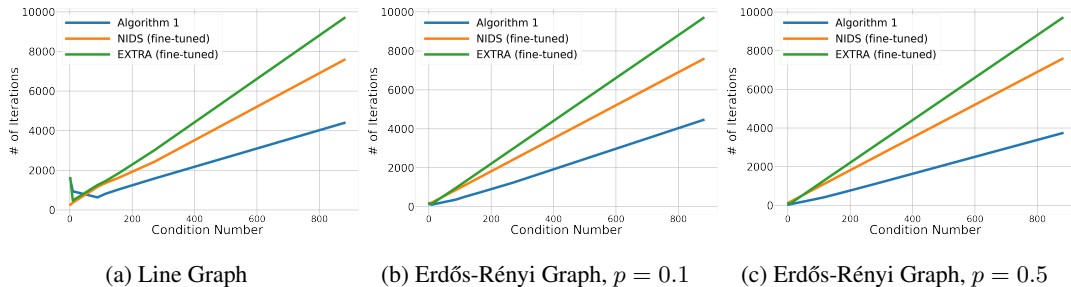

| (a) Line Graph | (b) Erdős-Rényi Graph, $p = 0.1$ | (c) Erdős-Rényi Graph, $p = 0.5$ |

Figure 2: **Ridge regression**: Number of iterations $N$ for $\|\mathbf{X}^N - \mathbf{X}^\star\| \leq 10^{-5}$ versus the condition number of agents' looses on different graphs; (2a) Line graph; (2b) Erdős-Rényi Graph with edge activation probability $p = 0.1$; (2c) Erdős-Rényi Graph with edge activation probability $p = 0.5$

The figures demonstrate that the proposed method consistently outperforms both EXTRA and NIDS, even when using the local min-consensus strategy, with a significant gap emerging as the condition number increases. This performance is particularly noteworthy given that Algorithm 1 and 3 operate effectively without requiring tedious tuning or global knowledge of the optimization and network parameters. Notably, Algorithms 1 and 3 exhibit different convergence behaviors: as predicted by Theorem 5, local min-consensus results in nonmonotonic error dynamics $\|\mathbf{X}^k - \mathbf{X}^\star\|^2$. However, the practical convergence speed remains largely unaffected compared to the global min-consensus.

## Acknowledgment

The work of I. Kuruzov has been supported by the Grant No. 70-2021-00143 01.11.2021, IGK 000000D730324P540002.

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

# Appendix

## A Proof of Lemma 3

The proof of the lemma is quite standard, and it is reported here for the sake of completeness.

**(i)** Smoothness of $f$ implies that Algorithm 2 terminates when $\alpha^+ \leq \delta/L_f$. Therefore, it must be $\alpha^+ \geq \min(\delta/2L_f, \gamma\alpha)$. Furthermore, it follows from the strong convexity of $f$ that $\delta/(2\alpha^+) \geq \mu/2$; hence, $\alpha^+ \leq \min(\delta/\mu, \gamma\alpha)$. This proves (8).

Further, by the lower bound above, one infers that the backtracking procedure terminates when $\alpha^+ \leq \frac{\delta}{L_i}$. Noting that $\alpha^+ = 2^{-t+1}\gamma$, we deduce that $t = \left\lfloor \log_2 \frac{2L_i\gamma\alpha}{\delta} \right\rfloor$ interations suffice.

**(ii)** Let $\phi(\alpha) := f(x + \alpha d)$. Notice that $\phi$ is convex and $\phi'(0) = \langle \nabla f(x), d \rangle$. The termination condition in Algorithm 2 can be equivalently rewritten in terms of $\phi$ as

$$\phi(\alpha^+) \leq \phi(0) + \phi'(0)\,\alpha^+ + \alpha^+ \frac{\delta}{2}\|d\|^2. \tag{15}$$

Given $\lambda \in [0, 1]$, let $\bar{\alpha} = \lambda\alpha^+$. Invoking convexity of $\phi$, we can write

$$\phi(\bar{\alpha}) = \phi\left(\lambda\alpha^+ + (1-\lambda)0\right) \leq \lambda\phi(\alpha^+) + (1-\lambda)\phi(0)$$
$$\overset{(15)}{\leq} \phi(0) + \phi'(0)\,(\lambda\alpha^+) + (\lambda\alpha^+)\frac{\delta}{2}\|d\|^2,$$

which completes the proof. $\qquad\qquad\square$

## B Proof of Theorem 4

We begin establishing the dynamics of $V^k$ defined in (10) along two consecutive updates.

**Lemma 6.** *The following holds along the update* $(\mathbf{D}^k, \mathbf{X}^k) \to (\mathbf{D}^{k+1}, \mathbf{X}^{k+1})$:

$$\begin{aligned}
V^{k+1} = &\left\|\mathbf{X}^k - \mathbf{X}^\star\right\|^2 + (\alpha^k)^2\|\mathbf{D}^k - \mathbf{D}^\star\|_M^2 \\
&- \|\mathbf{X}^k - \mathbf{X}^{k+1}\|^2 - (\alpha^k)^2\|\mathbf{D}^k - \mathbf{D}^{k+1}\|_M^2 \\
&+ 2\alpha^k \left\langle \nabla F(\mathbf{X}^{k+1/2}) - \nabla F(\mathbf{X}^\star), \mathbf{X}^\star - \mathbf{X}^{k+1} \right\rangle.
\end{aligned} \tag{16}$$

*Proof.* See Appendix E.1. $\qquad\qquad\square$

Using the properties of the backtracking procedure (Lemma 3) and leveraging strong convexity and smoothness of $F$, the inner product in (16) can be bounded as follows.

**Lemma 7.** *The following holds:*

$$\begin{aligned}
\left\langle \nabla F(\mathbf{X}^{k+1/2}) - \nabla F(\mathbf{X}^\star), \mathbf{X}^\star - \mathbf{X}^{k+1} \right\rangle \leq &\frac{\delta}{2\alpha^k}\|\mathbf{X}^{k+1} - \mathbf{X}^{k+1/2}\|^2 \\
&- \max\left(\frac{\mu^k}{2}\|\mathbf{X}^{k+1/2} - \mathbf{X}^\star\|^2, \frac{1}{2L^k}\|\nabla F(\mathbf{X}^{k+1/2}) + \mathbf{D}^*\|^2\right),
\end{aligned}$$

*where* $\mu^k$ *(resp.* $L^k$*) is the strong convexity (resp. smoothness) constant of each* $f_i$ *along the segment* $[x_i^{k+1/2}, x^\star]$.

*Proof.* See Appendix E.2. $\qquad\qquad\square$

Combining Lemma 6 and Lemma 7, after some algebraic manipulation, we obtain the following.

**Lemma 8.** *In the setting above, it holds*

$$\begin{aligned}
V^{k+1} \leq &\left\|\mathbf{X}^k - \mathbf{X}^\star\right\|^2 + (\alpha^k)^2\|\mathbf{D}^k - \mathbf{D}^\star\|_M^2 \\
&- \max\left(\mu^k\alpha^k\|\mathbf{X}^{k+1/2} - \mathbf{X}^\star\|^2, \frac{\alpha^k}{L^k}\|\nabla F(\mathbf{X}^{k+1/2}) + \mathbf{D}^*\|^2\right) \\
&- \delta\left(\|\mathbf{X}^k\|_{c(I-\widetilde{W})}^2 + (\alpha^k)^2\left\|c(I - \widetilde{W})\left(\nabla F(\mathbf{X}^{k+1/2}) + \mathbf{D}^k\right)\right\|_M^2\right).
\end{aligned} \tag{17}$$

*Proof.* See Appendix E.3. □

Lemma 8 suggests the path for the rest of the analysis: the decrease of $V^{k+1}$ relies on the values of the terms

$$\|\mathbf{X}^k\|^2_{c(I-\widetilde{W})} + (\alpha^k)^2 \left\|c(I - \widetilde{W})\left(\nabla F(\mathbf{X}^{k+1/2}) + \mathbf{D}^k\right)\right\|^2_M$$

and

$$\max\left(\mu^k\alpha^k\|\mathbf{X}^k - \mathbf{X}^\star\|^2, \frac{\alpha^k}{L}\|\nabla F(\mathbf{X}^{k+1/2}) + \mathbf{D}^*\|^2\right)$$

relative to the the primal and dual optimality gaps $\|\mathbf{X}^k - \mathbf{X}^\star\|^2$ and $(\alpha^k)^2\|\mathbf{D}^k - \mathbf{D}^\star\|^2_M$, respectively. At the high-level, one can say that "higher" values of such quantities relative to $\|\mathbf{X}^k - \mathbf{X}^\star\|^2$ and $(\alpha^k)^2\|\mathbf{D}^k - \mathbf{D}^\star\|^2_M$, determine larger decrease of $V^{k+1}$.

The above argument can be formally recorded by the following quantities:

$$r^k := \frac{\sqrt{\frac{1}{(\alpha^k)^2}\|\mathbf{X}^k\|^2_{c(I-\widetilde{W})} + \left\|c(I - \widetilde{W})\left(\nabla F(\mathbf{X}^{k+1/2}) + \mathbf{D}^k\right)\right\|^2_M}}{\max\left(\frac{1}{\alpha^k}\|\mathbf{X}^k - \mathbf{X}^\star\|, \|\mathbf{D}^k - \mathbf{D}^\star\|_M\right)}, \tag{18}$$

and

$$g^k := \frac{\max\left(\frac{1}{\alpha^k}\|\mathbf{X}^k - \mathbf{X}^\star\|, \|\nabla F(\mathbf{X}^{k+1/2}) + \mathbf{D}^*\|\right)}{\|\mathbf{D}^k - \mathbf{D}^\star\|_M}. \tag{19}$$

Using $r^k$ and $g^k$ in (17), the next lemma establishes contraction of $V^{k+1}$, with a contraction factor depending in particular on such quantities.

**Lemma 9.** *The following holds*

$$V^{k+1} \leq \max\left(\frac{\alpha^k}{\alpha^{k-1}}, 1\right)^2 \left(1 - \min\left(\rho_1^k, \zeta^k\right)\right) V^k, \tag{20}$$

*where*

$$\rho_1^k := \frac{\mu^k\alpha^k}{2}(1 - c(1 - \lambda_m(\widetilde{W})))^2 < 1$$

*and*

$$\zeta^k := \max\left(\delta(r^k)^2, (g^k)^2 \min\left(\frac{\mu^k\alpha^k(1 - c(1 - \lambda_m(\widetilde{W})))^2}{2}, \frac{1}{2L^k\alpha^k}\right)\right).$$

*Proof.* See Appendix E.4. □

The final expression (11) in Theorem 4 follows easily from $\zeta^k \geq \delta(r^k)^2$.

The above result ensures a "sufficient" descent of $V^{k+1}$ when $r^k$ (or $g^k$) is large enough. However, the contraction factor in (20) becomes vacuous for arbitrarily small values of $r^k$ (or $g^k$).

Next, we examine the unfavorable case where both $r^k$ *and* $g^k$ are "small", leading to the proof of the decay of $V^{k+1}$ as stated in (12) of Theorem 4. We build on the following key property of the sequence $g^k$ in this scenario: under low $r^k$ values, if $g^k$ is "small", the subsequent value $g^{k+1}$ cannot become arbitrarily small.

**Lemma 10.** *Suppose*

$$r^k < \frac{1}{\sqrt{2}} \quad and \quad g^k \leq \min\left(\frac{1 - r^k\sqrt{2}}{2\sqrt{\lambda_{\max}(M)}}, 1\right). \tag{21}$$

*Then,*

$$\frac{1}{(\alpha^k)^2}\frac{\|\mathbf{X}^{k+1} - \mathbf{X}^\star\|^2}{\|\mathbf{D}^{k+1} - \mathbf{D}^\star\|^2_M} \geq \frac{1}{\lambda_{\max}(M)}\left(1 - \frac{2g^k\sqrt{\lambda_{\max}(M)}}{1 - r^k\sqrt{2}}\right)^2. \tag{22}$$

*Proof.* See Appendix E.5. □

We infer from Lemma 10 that

$$g^{k+1} \overset{(19)}{\geq} \frac{1}{\alpha^{k+1}} \frac{\|\mathbf{X}^{k+1} - \mathbf{X}^\star\|}{\|\mathbf{D}^{k+1} - \mathbf{D}^\star\|_M}$$

$$\overset{(22)}{\geq} \frac{1}{\gamma^k \sqrt{\lambda_{\max}(M)}} \left( 1 - \frac{2g^k \sqrt{\lambda_{\max}(M)}}{1 - r^k \sqrt{2}} \right),$$

where we used $\alpha^k / \alpha^{k+1} \geq 1/\gamma^k$ (due to $\alpha^{k+1} \leq \gamma^k \alpha^k$, see Step 1 of Algorithm 2). Notice that (i) the term in the parenthesis will be around one for small enough values of $g^k$ and $r^k$; and (ii) the sequence $\{\gamma^k\}$ is chosen being eventually uniformly lower bounded. Therefore, the above bound implies that $r^k$ and $g^k$ cannot both progressively diminish along the iterates. Consequently, in the unfavorable scenario described by (21), $V^{k+1}$ still decreases, albeit over two consecutive iterations. This outcome is formalized in the following lemma.

**Lemma 11.** *Suppose condition* (21) *holds. Then,*

$$V^{k+2} \leq \max \left( \frac{\alpha^{k+1}}{\alpha^k}, 1 \right)^2 \max \left( \frac{\alpha^k}{\alpha^{k-1}}, 1 \right)^2 \left( 1 - \hat{\rho}_2^k \right) V^k, \tag{23}$$

*where*

$$\hat{\rho}_2^k := \frac{\mu^{k+1} \alpha^{k+1} \left( 1 - c(1 - \lambda_m(\widetilde{W})) \right)^2}{2(\gamma^k)^2 \max(\lambda_{\max}(M), 1)} \left( 1 - \frac{2g^k \sqrt{\lambda_{\max}(M)}}{1 - r^k \sqrt{2}} \right)^2 < 1.$$

*Here, $\mu^{k+1}$ is the strong convexity constants of (each) $f_i$ along the segment $[x_i^{(k+1)+1/2}, x^\star]$.*

*Proof.* See Appendix E.6. □

The final convergence result as stated in (12) is obtained using Lemma 9 and Lemma 11, with the variable $g^k$, absorbed, as outlined next. We strengthen condition on $r^k$ in (21) by $r^k \leq \sqrt{2}/4$. We consider two cases for the value of $g^k$, in the above scenario. Specifically, **(Case 1)** $g^k$ is bounded away from zero, implying $\zeta^k$ in (20) to be so. Therefore, $\left( 1 - \min \left( \rho_1^k, \zeta^k \right) \right) < 1$. This will be sufficient to ensure enough descent for $V^{k+1}$, though in two consecutive iterations. **(Case 2):** $g^k$ may be arbitrarily small; in this case, $\hat{\rho}_2^k$ in (23) remains bounded away from zero, ensuring contraction though from $V^k$ to $V^{k+2}$. More formally we have the following.

● **Case 1:** Consider (20), under (21) strengthened by $r^k \leq \sqrt{2}/4$. If

$$g^k \geq \min \left( \frac{1}{8\sqrt{\lambda_{\max}(M)}}, 1 \right), \tag{24}$$

then

$$\min(\rho_1^k, \zeta^k) \geq \frac{\left( 1 - c(1 - \lambda_m(\widetilde{W})) \right)^2}{16 \max(1, 8\lambda_{\max}(M))} \min \left( \mu^k \alpha^k, \frac{1}{\alpha^k L^k} \right).$$

Using lower bound above in (20), yields

$$V^{k+2} \leq \max \left( \frac{\alpha^{k+1}}{\alpha^k}, 1 \right)^2 V^{k+1} \leq \max \left( \frac{\alpha^{k+1}}{\alpha^k}, 1 \right)^2 \max \left( \frac{\alpha^k}{\alpha^{k-1}}, 1 \right)^2 \left( 1 - \hat{\rho}_1^k \right) V^k,$$

with

$$\hat{\rho}_1^k = \frac{\left( 1 - c(1 - \lambda_m(\widetilde{W})) \right)^2}{16 \max(1, 8\lambda_{\max}(M))} \min \left( \mu^k \alpha^k, \frac{1}{\alpha^k L^k} \right). \tag{25}$$

Notice that if the interval of admissible values for $g^k$, as specified by (21) and (24) is empty, this case does not apply.

● **Case 2:** Consider (23) under (21) strengthened by $r^k \leq \sqrt{2}/4$. If

$$0 < g^k \leq \min \left( 1, \frac{1}{8\sqrt{\lambda_{\max}(M)}} \right),$$

then
$$1 - \frac{2g^k\sqrt{\lambda_{\max}(M)}}{1 - r^k\sqrt{2}} > \frac{1}{2},$$
where we used $r^k \leq \sqrt{2}/4$. This implies
$$\hat{\rho}_2^k \geq \frac{\mu^{k+1}\alpha^{k+1}\left(1 - c(1 - \lambda_m(\widetilde{W}))\right)^2}{8(\gamma^k)^2\max(\lambda_{\max}(M), 1)}.$$
Using this lower bound in (23), yields
$$V^{k+2} \leq \max\left(\frac{\alpha^{k+1}}{\alpha^k}, 1\right)^2 \max\left(\frac{\alpha^k}{\alpha^{k-1}}, 1\right)^2 \left(1 - \hat{\rho}_3^k\right) V^k,$$
with
$$\hat{\rho}_3^k = \frac{\left(1 - c(1 - \lambda_m(\widetilde{W}))\right)^2}{8\max(\lambda_{\max}(M), 1)}\frac{\mu^{k+1}\alpha^{k+1}}{(\gamma^k)^2}. \tag{26}$$

Combining Case 1 and Case 2 above–taking the minimum between (25) and (26) and using the fact that $\gamma^k \geq 1$, leads to the desired decay of $V^{k+1}$ as in (12).

This completes the proof of Theorem 4. □

## C   Proof of Corollary 4.1

Let us consider the first $N > 0$ iterations of Algorithm 1. Let us denote by $L$ and $\mu$ the constants of smoothness and strong convexity of each $f_i$ restricted to the convex hull of $\{x^\star, \{x_i^k, x_i^{k+1/2}\}_{k=0}^N\}$. We proceed lower bounding $\rho_1^k$ and $\rho_2^k$ given in Theorem 4. We will use the following facts:
$$\alpha^k \geq \delta/(2L), \quad \alpha^k < \delta/\mu, \quad \text{and} \quad \frac{\alpha^{k+1}}{\alpha^k} \leq \gamma^k,$$
due to Lemma 7, and given $\lambda_{\max}(M) = (c(1 - \lambda_2(\widetilde{W}))^{-1} - 1$,
$$\frac{1}{\max(\lambda_{\max}(M), 1)} \geq c(1 - \lambda_2(\widetilde{W})) \quad \text{and} \quad \frac{1}{\lambda_{\max}(M)} \geq c(1 - \lambda_2(\widetilde{W})).$$
We can bound $\rho_1^k$ and $\rho_2^k$ as
$$\rho_1^k \geq \delta\frac{\mu}{4L}(1 - c(1 - \lambda_m(\widetilde{W})))^2 \quad \text{and} \quad \rho_2^k \geq \delta\frac{\mu}{L}\frac{\left(1 - c(1 - \lambda_m(\widetilde{W}))\right)^2 c(1 - \lambda_2(\widetilde{W}))}{256(\gamma^k)^2}, \quad \forall k \leq N. \tag{27}$$

Using (27), we can simplify the rate decay of $V^N$ in Theorem 4 as follows.

• **Case 1:** Suppose
$$r^k \geq r_{\text{low}} := \frac{1}{\sqrt{2}}\min\left(\frac{1}{2}, \frac{1}{\sqrt{\lambda_{\max}(M)}}\right), \quad \forall k \geq N. \tag{28}$$
Substituting the lower bounds of $\rho_1^k$ and $r^k$ in (11), we obtain the following simplified convergence rate:
$$V^N \leq \left(\prod_{k=0}^{N-1}\gamma^k\right)^2 \left(1 - \frac{\delta}{8}\min\left(\frac{\mu}{L}\left(1 - c(1 - \lambda_m(\widetilde{W}))\right)^2, c(1 - \lambda_2(\widetilde{W}))\right)\right)^N V^0.$$

• **Case 2:** Condition (28) does not hold. For the values of $k$ such that $r^k \leq \frac{1}{\sqrt{2}}\min\left(\frac{1}{2}, \frac{1}{\sqrt{\lambda_{\max}(M)}}\right) \leq \frac{\sqrt{2}}{4}$, we can use (12). Substituting threin the lower bound for $\rho_2$ and $\gamma^k = ((k + \beta_1)/(k + 1))^{\beta_2} \geq \beta_1^{\beta_2}$, yields
$$V^{k+2} \leq \left(\gamma^k\gamma^{k+1}\right)^2 \left(1 - \delta\frac{(1 - c(1 - \lambda_m(\widetilde{W})))^2 c(1 - \lambda_2(\widetilde{W}))}{256\beta_1^{2\beta_2}}\frac{\mu}{L}\right) V^k. \tag{29}$$

On the other hand, for $k$ such that $r^k \geq \frac{1}{\sqrt{2}} \min\left(\frac{1}{2}, \frac{1}{\sqrt{\lambda_{\max}(M)}}\right)$, using (11) on two consecutive iterations, we have

$$
\begin{aligned}
V^{k+2} &\leq (\gamma^{k+1})^2 V^{k+1} \\
&\leq \left(\gamma^k \gamma^{k+1}\right)^2 \left(1 - \frac{\delta}{8} \min\left(\frac{\mu}{L}\left(1 - c(1 - \lambda_m(\widetilde{W}))\right)^2, c(1 - \lambda_2(\widetilde{W}))\right)\right) V^k \\
&\leq \left(\gamma^k \gamma^{k+1}\right)^2 \left(1 - \delta \frac{(1 - c(1 - \lambda_m(\widetilde{W})))^2 c(1 - \lambda_2(\widetilde{W}))}{256\beta_1^{2\beta_2}} \frac{\mu}{L}\right) V^k.
\end{aligned}
\tag{30}
$$

Therefore, in either situations of Case 2, one can ensure contraction after two consecutive iterations by a factor

$$
\delta \frac{(1 - c(1 - \lambda_m(\widetilde{W})))^2 c(1 - \lambda_2(\widetilde{W}))}{256\beta_1^{2\beta_2}} \frac{\mu}{L} < 1.
$$

Using $V^{k+1} \leq (\gamma^k)^2 V^k$, we can merge (29) and (30) as follows:

$$
V^N \leq \left(\prod_{k=0}^{N-1} \gamma^k\right)^2 \left(1 - \delta \frac{(1 - c(1 - \lambda_m(\widetilde{W})))^2 c(1 - \lambda_2(\widetilde{W}))}{256\beta_1^{2\beta_2}} \frac{\mu}{L}\right)^{\lfloor N/2 \rfloor} V^0
$$

$$
\leq \left(\prod_{k=0}^{N-1} \gamma^k\right)^2 \left(1 - \delta \frac{(1 - c(1 - \lambda_m(\widetilde{W})))^2 c(1 - \lambda_2(\widetilde{W}))}{256\beta_1^{2\beta_2}} \frac{\mu}{L}\right)^{(N-1)/2} V^0
$$

$$
\leq \left(\prod_{k=0}^{N-1} \gamma^k\right)^2 \left(1 - \delta \frac{(1 - c(1 - \lambda_m(\widetilde{W})))^2 c(1 - \lambda_2(\widetilde{W}))}{512\beta_1^{2\beta_2}} \frac{\mu}{L}\right)^{N-1} V^0.
$$

• **Case 1 + Case 2:** We can combine the rate expressions derived in the two cases above as follows:

$$
V^N \leq \left(\prod_{k=0}^{N-1} \gamma^k\right)^2 (1 - \rho)^{N-1} V^0,
$$

where

$$
\rho = \begin{cases}
\frac{\delta}{8} \min\left(\frac{\mu}{4L}(1 - c(1 - \lambda_m(\widetilde{W})))^2, c(1 - \lambda_2(\widetilde{W}))\right), & \text{if } r^k \geq \frac{1}{\sqrt{2}} \min\left(\frac{1}{2}, \frac{1}{\sqrt{\lambda_{\max}(M)}}\right) \text{ for all } k; \\
\delta \frac{(1 - c(1 - \lambda_m(\widetilde{W})))^2 c(1 - \lambda_2(\widetilde{W}))}{512(\gamma^k)^2} \frac{\mu}{L}, & \text{else.}
\end{cases}
$$

Notice that $\rho \in (0, 1)$.

Finally, we can obtain the desired asymptotic convergent rate noting that the growth of $\prod_k \gamma^k$ is dominated by the geometric decay of the contraction factor. This is formalized next.

**Lemma 12.** *Let $\gamma^k = ((k + \beta_1)/(k + 1))^{\beta_2}$ with $\beta_1 \geq 1, \beta_2 \geq 0$. Then the following holds:*

$$
\prod_{k=0}^{N-1} \gamma^k \leq \beta_1^{\beta_2(\lceil \beta_1 \rceil + 1)} N^{\beta_2(\beta_1 - 1)}.
\tag{31}
$$

*Furthermore, for any given $\rho \in (0, 1)$, we have*

$$
\left(\prod_{k=0}^{N-1} \gamma^k\right)^2 (1 - \rho)^{N-1} \leq (1 - \rho/2)^{N-1},
\tag{32}
$$

*for all*

$$
N \geq N_0 := \frac{4}{\rho} \max\left(2\beta_2(\lceil \beta_1 \rceil + 1) \ln \beta_1 + \ln(2), 4\beta_2 \beta_1 \ln \frac{8\beta_1\beta_2}{\rho}\right).
$$

*Proof.* See Appendix E.7 □

Inequality (32) provides the asymptotic rate expression, as stated in the corollary where $\mathcal{O}$ hides the dependence on $\beta_1$ and $\beta_2$. □

# D   Proof of Theorem 5

We begin noticing that if the stepsizes in Algorithm 3 are identical across agents, Algorithm 3 reduces to Algorithm 1. For the iterates where this happens, one can rely on the convergence guarantees established for Algorithm 1. Specifically, we have the following result, whose proof is straightforward.

**Lemma 13.** *Suppose that exists some $k_\varepsilon \geq 1$ such that $\alpha_1^k = \cdots = \alpha_m^k$, for $k = k_\varepsilon, \ldots, k_\varepsilon + N_\varepsilon$. Then one can guarantee*

$$\left\| \mathbf{X}^{k_\varepsilon + N_\varepsilon} - \mathbf{X}^\star \right\|^2 + \frac{1}{4L^2} \| \mathbf{D}^{K_\varepsilon + N_\varepsilon} - \mathbf{D}^\star \|_M^2 \leq \varepsilon,$$

*where $N_\varepsilon$ is defined as in Corollary 4.1 (replacing therein $V^0$ with $V^{K_\varepsilon}$).*

The remainder of the proof focuses on characterizing the properties of certain key events identified as detrimental for the local-min consensus algorithm to achieve convergence. We will demonstrate that the occurrence of such events within $N$ consecutive iterations is of the order of $\log N$, indicating that these are sporadic events relative to the total of $N$ iterations.

Given $\alpha_i^k$'s, as defined in Step (S.3) of Algorithm 3, let us denote their minimum across *all* agents at time $k$ as

$$\alpha_{\min}^k := \min_{i \in [m]} \alpha_i^k.$$

If the backtracking loop (steps 3-6 in Algorithm 2) is not activated at iteration $k - 1$ in *any* of the agents' local line searches, then all output stepsizes $\alpha_i^k$ will increase by the same factor $\gamma^k$, that is, $\alpha_i^k = \gamma^k \alpha_i^{k-1}$; hence, does $\alpha_{\min}^k$. On the other hand, if all stepsizes are consensual at iteration $k - 1$ and the backtracking procedure at *some* of the agents' side enters its steps- 3-6, a "desynchronization" of the stepsizes occurs. This event can be detected by the condition

$$\alpha_{\min}^k < \gamma^k \alpha_{\min}^{k-1}.$$

When the stepsize at time $k - 1$ are not consensual, the condition above identifies increases in stepsize disagreements from iteration $k - 1$ to $k$, measured by

$$\max_{j \in [m]} \alpha_j^k - \alpha_{\min}^k > \gamma^k \left( \max_{j \in [m]} \alpha_j^{k-1} - \alpha_{\min}^{k-1} \right).$$

This motivates the definition of the following index set: given $N = 1, 2, \ldots$, let

$$\mathcal{I}_N = \left\{ k \in [N] : \alpha_{\min}^k < \gamma^k \alpha_{\min}^{k-1} \right\}.$$

The key properties of interest of this set are summarized below.

**Lemma 14.** *For any given $N = 1, 2, \ldots$, the following statements hold:*

1. *If $\alpha_i^k = \alpha_{\min}^k$, for all $i \in [m]$, and $k + 1 \notin \mathcal{I}_N$, then $\alpha_i^{k+1} = \alpha_{\min}^{k+1}$, for all $i \in [m]$;*

2. *If $k \in \mathcal{I}_N$, $k < N - d_\mathcal{G}$, and $k + 1, \ldots k + d_\mathcal{G} \notin \mathcal{I}_N$, then $\alpha_i^{k+d_\mathcal{G}} = \alpha_{\min}^{k+d_\mathcal{G}}$, for all $i \in [m]$;*

3. *$|\mathcal{I}_N| \leq \max \left( \ln \alpha_0 L + \ln \prod_{k=0}^{N-1} \gamma^k + \ln \frac{2}{\delta}, 0 \right).$*

*Proof.* See Appendix F.1. □

In words, the first statement confirms that consensus on the stepsizes is maintained if none of the local backtracking procedures are triggered. The second statement ensures that, the local-min consensus algorithm requires at most $d_\mathcal{G}$ iterations to converge, from any initialization, provided that during those iterations no backtracking events alter the minimum stepsize across agents. Lastly, the third

assertion provides a limit on the maximum number of detrimental events that can occur during the $N$ iterations under consideration. If this number is small relative to $N$, a fact that will be proved shortly, one we can find (multiple) window(s) of consecutive iterations wherein the stepsizes remain consensual across all agents. Within these windows, Lemma 13 can be applied, to establish convergence. This idea is formalized next.

**Lemma 15.** *Suppose $V^k \leq R$. Then,*

$$\min_{j \in [1, N+1]} \left\| \mathbf{X}^j - \mathbf{X}^\star \right\|^2 + \frac{1}{4L^2} \|\mathbf{D}^j - \mathbf{D}^\star\|_M^2 \leq \varepsilon, \tag{33}$$

*if*

$$\frac{N}{N_\varepsilon + d_{\mathcal{G}}} > |\mathcal{I}_N| + 1. \tag{34}$$

*Here, $N_\varepsilon$ is defined as in Corollary 4.1 (replacing therein $V_0$ with $R$).*

*Proof.* See Appendix F.2. $\qquad\square$

To finalize our proof, let us simplify an upper bound of $|\mathcal{I}_N|$, when $\gamma^k = \left( \frac{k+\beta_1}{k+1} \right)^{\beta_2}$. For the sake of simplicity, we consider the case $\ln N \geq 1$. Invoking Lemma 12, we have

$$\ln \frac{2}{\delta} + \ln \prod_{k=0}^{N-1} \gamma^k \overset{(31)}{\leq} \ln \frac{2}{\delta} + \beta_2(\lfloor \beta_1 \rfloor + 1) \ln \beta_1 + \beta_2(\beta_1 - 1) \ln N$$

$$\leq \underbrace{\left( \beta_2(\lfloor \beta_1 \rfloor + 1) \ln \beta_1 + \beta_2(\beta_1 - 1) + \ln \ln \frac{2}{\delta} \right)}_{\xi :=} \ln N$$

$$= \xi \ln N,$$

where the constant $\xi$ depends on the algorithm parameters. Therefore, one can guarantee (33), under the following condition

$$\frac{N}{\xi \ln N + \ln \alpha_0 L} \geq N_\varepsilon + d_{\mathcal{G}}. \tag{35}$$

A sufficient condition for (35) is

$$\frac{N}{\max \left( \ln N, \ln \alpha_0 L \right)} \geq 2 \max(\xi, 1) \max(N_\varepsilon, d_{\mathcal{G}}).$$

Let $N^*$ be the smallest iteration for which the above inequality holds. Then,

$$N^* = \mathcal{O}(\max \left[ \log d_{\mathcal{G}} + \log N_\varepsilon, \log \alpha_0 L \right] \max(N_\varepsilon, d_{\mathcal{G}})).$$

$\qquad\square$

# E   Proof of the Intermediate Results in Appendix B

## E.1   Proof of Lemma 6

Let us rewrite $V^{k+1}$ in terms of $\|\mathbf{X}^k - \mathbf{X}^\star\|^2$ and $(\alpha^k)^2 \|\mathbf{D}^k - \mathbf{D}^\star\|_M^2$, to link it back to $V^k$:

$$\begin{aligned}
V^{k+1} = & \|\mathbf{X}^k - \mathbf{X}^\star\|^2 + (\alpha^k)^2 \|\mathbf{D}^k - \mathbf{D}^\star\|_M^2 \\
& - \|\mathbf{X}^k - \mathbf{X}^{k+1}\|^2 - (\alpha^k)^2 \|\mathbf{D}^k - \mathbf{D}^{k+1}\|_M^2 \\
& - 2 \underbrace{\left\langle \mathbf{X}^{k+1} - \mathbf{X}^\star, \mathbf{X}^k - \mathbf{X}^{k+1} \right\rangle}_{\texttt{term I}} - 2 (\alpha^k)^2 \underbrace{\left\langle \mathbf{D}^{k+1} - \mathbf{D}^\star, \mathbf{D}^k - \mathbf{D}^{k+1} \right\rangle_M}_{\texttt{term II}}.
\end{aligned} \tag{36}$$

where the equality follows from $\|a\|^2 = \|b\|^2 - \|a - b\|^2 - 2\langle a, b - a \rangle$.

Notice that the negative terms on the RHS of (36) will contribute to the decrease of $V^{k+1}$. We are thus left to deal with `term I` and `term II`. The idea is to bound them so that they can overall being controlled by the backtracking inequality.

Let us proceed bounding `term I` and `term II` using the algorithm dynamics. We have the following:

$$
\begin{aligned}
\texttt{term I} &= \left\langle (\mathbf{X}^k - \alpha^k \mathbf{D}^{k+1}) - \mathbf{X}^\star, \mathbf{X}^k - \mathbf{X}^{k+1} \right\rangle - \alpha^k \left\langle \nabla F(\mathbf{X}^{k+1/2}), \mathbf{X}^k - \mathbf{X}^{k+1} \right\rangle \\
&= \left\langle \mathbf{X}^k - \mathbf{X}^\star, \mathbf{X}^k - \mathbf{X}^{k+1} \right\rangle + \alpha^k \left\langle \mathbf{D}^{k+1} - \mathbf{D}^\star, \mathbf{X}^{k+1} - \mathbf{X}^k \right\rangle \\
&\quad - \alpha^k \left\langle \nabla F(\mathbf{X}^{k+1/2}) - \nabla F(\mathbf{X}^\star), \mathbf{X}^k - \mathbf{X}^{k+1} \right\rangle,
\end{aligned}
\tag{37}
$$

where in the second equality we used $\nabla F(\mathbf{X}^\star) = -\mathbf{D}^\star$. Note that the last term in the expression above can be controlled through the backtracking procedure. Hence, we proceed bounding `term II` to "cancel" out the other terms on the RHS of (37). Specifically,

$$
\begin{aligned}
\texttt{term II} &= -\alpha^k \left\langle \mathbf{D}^{k+1} - \mathbf{D}^\star, \alpha^k c^{-1} \left( I - \widetilde{W} \right)^\dagger (\mathbf{D}^{k+1} - \mathbf{D}^k) + \alpha^k \mathbf{D}^k - \alpha^k \mathbf{D}^{k+1} \right\rangle \\
&= -\alpha^k \left\langle \mathbf{D}^{k+1} - \mathbf{D}^\star, \mathbf{X}^k - \alpha^k \nabla F(\mathbf{X}^{k+1/2}) - \alpha^k \mathbf{D}^{k+1} - \mathbf{X}^\star \right\rangle \\
&= -\alpha^k \left\langle \mathbf{D}^{k+1} - \mathbf{D}^\star, \mathbf{X}^{k+1} - \mathbf{X}^\star \right\rangle,
\end{aligned}
\tag{38}
$$

where we used the update of $\mathbf{X}^{k+1}$ and the facts $\mathbf{D}^{k+1} - \mathbf{D}^\star \in \mathtt{span}(I - \widetilde{W})$ and $\mathbf{X}^\star \in \mathtt{null}(I - \widetilde{W})$.

Summing up (37) and (38), we obtain

$$
\begin{aligned}
\texttt{term I} + \texttt{term II} &= \left\langle \mathbf{X}^k - \mathbf{X}^\star, \mathbf{X}^k - \mathbf{X}^{k+1} \right\rangle + \alpha^k \left\langle \mathbf{D}^{k+1} - \mathbf{D}^\star, \mathbf{X}^{k+1} - \mathbf{X}^k \right\rangle \\
&\quad - \alpha^k \left\langle \nabla F(\mathbf{X}^{k+1/2}) - \nabla F(\mathbf{X}^\star), \mathbf{X}^k - \mathbf{X}^{k+1} \right\rangle \\
&\quad - \alpha^k \left\langle \mathbf{D}^{k+1} - \mathbf{D}^\star, \mathbf{X}^{k+1} - \mathbf{X}^\star \right\rangle \\
&= \left\langle \mathbf{X}^k - \mathbf{X}^\star, \mathbf{X}^k - \mathbf{X}^{k+1} \right\rangle - \alpha^k \left\langle \mathbf{D}^{k+1} - \mathbf{D}^\star, \mathbf{X}^k - \mathbf{X}^\star \right\rangle \\
&\quad - \alpha^k \left\langle \nabla F(\mathbf{X}^{k+1/2}) - \nabla F(\mathbf{X}^\star), \mathbf{X}^k - \mathbf{X}^{k+1} \right\rangle \\
&= \alpha^k \left\langle \mathbf{X}^k - \mathbf{X}^\star, \frac{1}{\alpha^k} \left( \mathbf{X}^k - \mathbf{X}^{k+1} \right) - \mathbf{D}^{k+1} + \mathbf{D}^\star \right\rangle \\
&\quad - \alpha^k \left\langle \nabla F(\mathbf{X}^{k+1/2}) - \nabla F(\mathbf{X}^\star), \mathbf{X}^k - \mathbf{X}^{k+1} \right\rangle \\
&= \alpha^k \left\langle \mathbf{X}^k - \mathbf{X}^\star, \nabla F(\mathbf{X}^{k+1/2}) - \nabla F(\mathbf{X}^\star) \right\rangle \\
&\quad - \alpha^k \left\langle \nabla F(\mathbf{X}^{k+1/2}) - \nabla F(\mathbf{X}^\star), \mathbf{X}^k - \mathbf{X}^{k+1} \right\rangle \\
&= -\alpha^k \left\langle \nabla F(\mathbf{X}^{k+1/2}) - \nabla F(\mathbf{X}^\star), \mathbf{X}^\star - \mathbf{X}^{k+1} \right\rangle.
\end{aligned}
\tag{39}
$$

The statement of the Lemma follows readily substituting (39) in (36). $\qquad\square$

### E.2 Proof of Lemma 7

We preliminary notice that, in view of Lemma 3, the backtracking inequality (6) on $F^k$ holds with $\alpha^k = \min_{i \in [m]} \alpha_i^k$, where each $\alpha_i^k$ is the outcome of the backtracking procedure on the local $f_i^k$. Since $F$ and $F^k$ have the same curvature, it follows that (6) holds also on $F$, that is,

$$
F(\mathbf{X}^{k+1}) \leq F(\mathbf{X}^{k+1/2}) + \left\langle \nabla F(\mathbf{X}^{k+1/2}), \mathbf{X}^{k+1} - \mathbf{X}^{k+1/2} \right\rangle + \frac{\delta}{2\alpha^k} \|\mathbf{X}^{k+1} - \mathbf{X}^{k+1/2}\|^2. \tag{40}
$$

We proceed now bounding $\left\langle \nabla F(\mathbf{X}^{k+1/2}) - \nabla F(\mathbf{X}^\star), \mathbf{X}^\star - \mathbf{X}^{k+1} \right\rangle$ building on (40). To do so, we decompose the inner product as follows

$$
\left\langle \nabla F(\mathbf{X}^{k+1/2}) - \nabla F(\mathbf{X}^\star), \mathbf{X}^\star - \mathbf{X}^{k+1} \right\rangle
$$
$$
= \underbrace{\left\langle \nabla F(\mathbf{X}^{k+1/2}), \mathbf{X}^\star - \mathbf{X}^{k+1/2} \right\rangle}_{\texttt{term I}} - \underbrace{\left\langle \nabla F(\mathbf{X}^{k+1/2}), \mathbf{X}^{k+1} - \mathbf{X}^{k+1/2} \right\rangle}_{\texttt{term II}} + \left\langle \nabla F(\mathbf{X}^\star), \mathbf{X}^{k+1} - \mathbf{X}^\star \right\rangle
$$

$$\tag{41}$$

We bound `term I` invoking strong convexity and co-coercivity of $F$ while we use (40) to bound `term II`. Specifically,

$$\texttt{term I} \leq F(\mathbf{X}^\star) - F(\mathbf{X}^{k+1/2}) + \begin{cases} -\dfrac{\mu^k}{2}\left\|\mathbf{X}^{k+1/2} - \mathbf{X}^\star\right\|^2, & \text{(by strong convexity)} \\[2ex] -\dfrac{1}{2L^k}\left\|\nabla F(\mathbf{X}^{k+1/2}) + \mathbf{D}^\star\right\|^2, & \text{(by co-coercivity)}, \end{cases}$$

where we also used $\nabla F(\mathbf{X}^\star) = -\mathbf{D}^\star$. Therefore

$$\texttt{term I} \leq F(\mathbf{X}^\star) - F(\mathbf{X}^{k+1/2}) - \max\left(\frac{\mu^k}{2}\left\|\mathbf{X}^{k+1/2} - \mathbf{X}^\star\right\|^2, \frac{1}{2L^k}\left\|\nabla F(\mathbf{X}^{k+1/2}) + \mathbf{D}^\star\right\|^2\right).$$
$$(42)$$

Using (40), `term II` can be bounded as

$$\texttt{term II} \leq F(\mathbf{X}^{k+1/2}) - F(\mathbf{X}^{k+1}) + \frac{\delta}{2\alpha^k}\left\|\mathbf{X}^{k+1} - \mathbf{X}^{k+1/2}\right\|^2. \qquad (43)$$

Using (42) and (43) in (41), yields

$$\left\langle \nabla F(\mathbf{X}^{k+1/2}) - \nabla F(\mathbf{X}^\star), \mathbf{X}^\star - \mathbf{X}^{k+1}\right\rangle$$
$$\leq \frac{\delta}{2\alpha^k}\left\|\mathbf{X}^{k+1} - \mathbf{X}^{k+1/2}\right\|^2 - \max\left(\frac{\mu^k}{2}\left\|\mathbf{X}^{k+1/2} - \mathbf{X}^\star\right\|^2, \frac{1}{2L^k}\left\|\nabla F(\mathbf{X}^{k+1/2}) + \mathbf{D}^\star\right\|^2\right)$$
$$+ \underbrace{F(\mathbf{X}^\star) + \left\langle \nabla F(\mathbf{X}^\star), \mathbf{X}^{k+1} - \mathbf{X}^\star\right\rangle - F(\mathbf{X}^{k+1})}_{\leq 0}.$$

This completes the proof. $\qquad\square$

### E.3 Proof of Lemma 8

Combining Lemma 6 and Lemma 7, we can write

$$V^{k+1} \leq \left\|\mathbf{X}^k - \mathbf{X}^\star\right\|^2 + (\alpha^k)^2\|\mathbf{D}^k - \mathbf{D}^\star\|_M^2$$
$$- \max\left(\frac{\mu^k}{2}\|\mathbf{X}_k^{k+1/2} - \mathbf{X}^\star\|^2, \frac{1}{2L^k}\|\nabla F(\mathbf{X}^{k+1/2}) + \mathbf{D}^*\|^2\right)$$
$$- \underbrace{\|\mathbf{X}^k - \mathbf{X}^{k+1}\|^2}_{\texttt{term I}} - \underbrace{(\alpha^k)^2\|\mathbf{D}^k - \mathbf{D}^{k+1}\|_M^2}_{\texttt{term II}} + \delta\underbrace{\|\mathbf{X}^{k+1} - \mathbf{X}^{k+1/2}\|^2}_{\texttt{term III}}.$$

Next, we demonstrate that the sum of the last three terms contributes to the decrease of $V^{k+1}$.

Using the definition of $\mathbf{X}^{k+1}$ and $\mathbf{X}^{k+1/2}$, we can bound `term III` as follows:

$$\texttt{term III} = \left\|\left(\mathbf{X}^{k+1} - \mathbf{X}^k\right) - \left(\mathbf{X}^{k+1/2} - \mathbf{X}^k\right)\right\|^2$$
$$= -\left\|\mathbf{X}^{k+1/2} - \mathbf{X}^k\right\|^2 - 2\left\langle\mathbf{X}^{k+1/2} - \mathbf{X}^k, \mathbf{X}^{k+1} - \mathbf{X}^{k+1/2}\right\rangle + \left\|\mathbf{X}^{k+1} - \mathbf{X}^k\right\|^2$$
$$= -\left\|c(I - \widetilde{W})\mathbf{X}^k\right\|^2 - 2\alpha^k\left\langle\mathbf{D}^{k+1/2}, c(I - \widetilde{W})\mathbf{X}^k\right\rangle + \left\|\mathbf{X}^{k+1} - \mathbf{X}^k\right\|^2$$
$$= -\left\|c(I - \widetilde{W})\mathbf{X}^k\right\|^2 - 2\alpha^k\left\langle\left(I - c(I - \widetilde{W})\right)\left(\nabla F(\mathbf{X}^{k+1/2}) + \mathbf{D}^k\right), c(I - \widetilde{W})\mathbf{X}^k\right\rangle$$
$$+ \|\mathbf{X}^k - \mathbf{X}^{k+1}\|^2.$$

Proceeding with $\|\mathbf{D}^k - \mathbf{D}^{k+1}\|_M^2$, we have

$$\texttt{term II} = (\alpha^k)^2\left\|-c(I - \widetilde{W})\left(\nabla F(\mathbf{X}^{k+1/2}) + \mathbf{D}^k\right) + \frac{c}{\alpha^k}(I - \widetilde{W})\mathbf{X}^k\right\|_M^2$$
$$= -\left\|c(I - \widetilde{W})\mathbf{X}^k\right\|^2 + \|\mathbf{X}^k\|_{c(I-\widetilde{W})}^2$$
$$- 2\alpha^k\left\langle\left(I - c(I - \widetilde{W})\right)\left(\nabla F(\mathbf{X}^{k+1/2}) + \mathbf{D}^k\right), c(I - \widetilde{W})\mathbf{X}^k\right\rangle$$
$$+ (\alpha^k)^2\left\|c(I - \widetilde{W})\left(\nabla F(\mathbf{X}^{k+1/2}) + \mathbf{D}^k\right)\right\|_M^2,$$

where the second equality follows from the definition of $M = c^{-1}(I - \widetilde{W})^\dagger - I$.

Combining the three terms yields

$$
\begin{aligned}
&- \texttt{term I} - \texttt{term II} + \delta \texttt{term III} \\
={}& \delta(-\texttt{term I} - \texttt{term II} + \texttt{term III}) - (1 - \delta)(\texttt{term I} + \texttt{term II}) \\
\leq{}& \delta(-\texttt{term I} - \texttt{term II} + \texttt{term III}) \\
={}& - \delta \|\mathbf{X}^k\|^2_{c(I-\widetilde{W})} - \delta(\alpha^k)^2 \left\| c(I - \widetilde{W}) \left( \nabla F(\mathbf{X}^{k+1/2}) + \mathbf{D}^k \right) \right\|^2_M,
\end{aligned}
$$

which proves the statement of the lemma. $\qquad\qquad\qquad\qquad\qquad\qquad\qquad\square$

### E.4 Proof of Lemma 9

The proof involves further bounding the RHS of (17) in Lemma 8, appropriately in terms of $r^k$ and $g^k$. To achieve this, we construct two alternative bounds of the RHS of (17), as discussed below.

The first bound will reveal the dependence on $r^k$, and it is based on using in (17)

$$
\max\left( \mu^k \alpha^k \|\mathbf{X}^{k+1/2} - \mathbf{X}^\star\|^2, \frac{\alpha^k}{L^k} \|\nabla F(\mathbf{X}^{k+1/2}) + \mathbf{D}^*\|^2 \right) \geq \frac{\mu^k \alpha^k}{2} \|\mathbf{X}^{k+1/2} - \mathbf{X}^\star\|^2
$$

along with

$$
\|\mathbf{X}^{k+1/2} - \mathbf{X}^\star\|^2 \geq \left( 1 - c(1 - \lambda_m(\widetilde{W})) \right)^2 \|\mathbf{X}^k - \mathbf{X}^\star\|^2. \tag{44}
$$

We obtain

$$
\begin{aligned}
V^{k+1} \leq{}& \left\|\mathbf{X}^k - \mathbf{X}^\star\right\|^2 + (\alpha^k)^2 \|\mathbf{D}^k - \mathbf{D}^\star\|^2_M \\
&- \frac{\mu^k \alpha^k}{2} (1 - c(1 - \lambda_m(\widetilde{W})))^2 \|\mathbf{X}^k - \mathbf{X}^\star\|^2 \\
&- \delta \left\|\mathbf{X}^k\right\|^2_{c(I-\widetilde{W})} - \delta(\alpha^k)^2 \left\| c(I - \widetilde{W}) \left( \nabla F(\mathbf{X}^{k+1/2}) + \mathbf{D}^k \right) \right\|^2_M \\
\overset{(18)}{\leq}{}& \left\|\mathbf{X}^k - \mathbf{X}^\star\right\|^2 + (\alpha^k)^2 \|\mathbf{D}^k - \mathbf{D}^\star\|^2_M \\
&- \frac{\mu^k \alpha^k}{2} (1 - c(1 - \lambda_m(\widetilde{W})))^2 \|\mathbf{X}^k - \mathbf{X}^\star\|^2 \\
&- \delta(r^k)^2 (\alpha^k)^2 \|\mathbf{D}^k - \mathbf{D}^*\|^2_M \\
={}& \left( 1 - \frac{\mu^k \alpha^k}{2} (1 - c(1 - \lambda_m(\widetilde{W})))^2 \right) \|\mathbf{X}^k - \mathbf{X}^\star\|^2 + \left( 1 - \delta(r^k)^2 \right) (\alpha^k)^2 \|\mathbf{D}^k - \mathbf{D}^\star\|^2_M.
\end{aligned}
$$

The second bound of the RHS of (17) aims to obtain an explicit dependence on $g^k$. This is done by just neglecting the last two negative terms in the RHS of (17):

$$
\begin{aligned}
V^{k+1} \leq{}& \left\|\mathbf{X}^k - \mathbf{X}^\star\right\|^2 + (\alpha^k)^2 \|\mathbf{D}^k - \mathbf{D}^\star\|^2_M \\
&- \max\left( \mu^k \alpha^k \|\mathbf{X}^{k+1/2} - \mathbf{X}^\star\|^2, \frac{\alpha^k}{L^k} \|\nabla F(\mathbf{X}^{k+1/2}) + \mathbf{D}^*\|^2 \right) \\
\overset{(44)}{\leq}{}& \left\|\mathbf{X}^k - \mathbf{X}^\star\right\|^2 + (\alpha^k)^2 \|\mathbf{D}^k - \mathbf{D}^\star\|^2_M \\
&- \max\left( \mu^k \alpha^k (1 - c(1 - \lambda_m(\widetilde{W})))^2 \|\mathbf{X}^k - \mathbf{X}^\star\|^2, \frac{\alpha^k}{L^k} \|\nabla F(\mathbf{X}^{k+1/2}) + \mathbf{D}^*\|^2 \right) \\
\leq{}& \left\|\mathbf{X}^k - \mathbf{X}^\star\right\|^2 + (\alpha^k)^2 \|\mathbf{D}^k - \mathbf{D}^\star\|^2_M - \frac{\mu^k \alpha^k}{2} (1 - c(1 - \lambda_m(\widetilde{W})))^2 \|\mathbf{X}^k - \mathbf{X}^\star\|^2 \\
&- \min\left( \frac{\mu^k (\alpha^k)^3 (1 - c(1 - \lambda_m(\widetilde{W})))^2}{2}, \frac{\alpha^k}{2L^k} \right) \max\left( \frac{1}{(\alpha^k)^2} \|\mathbf{X}^k - \mathbf{X}^\star\|^2, \|\nabla F(\mathbf{X}^{k+1/2}) + \mathbf{D}^*\|^2 \right)
\end{aligned}
$$

$$= \left(1 - \frac{\mu^k \alpha^k}{2}(1 - c(1 - \lambda_m(\widetilde{W})))^2\right) \|\mathbf{X}^k - \mathbf{X}^\star\|^2$$
$$+ \left(1 - (g^k)^2 \min\left(\frac{\mu^k \alpha^k(1 - c(1 - \lambda_m(\widetilde{W})))^2}{2}, \frac{1}{2L^k\alpha^k}\right)\right)(\alpha^k)^2\|\mathbf{D}^k - \mathbf{D}^\star\|_M^2.$$

The final result follows combining the above two bounds while using the definition of $\rho_1^k$ and $\zeta^k$. $\square$

### E.5   Proof of Lemma 10

Invoking the update of the primal variable in the form

$$\mathbf{X}^{k+1} = \mathbf{X}^k - \alpha^k(\nabla F(\mathbf{X}^{k+1/2}) + \mathbf{D}^\star) - \alpha^k(\mathbf{D}^{k+1} - \mathbf{D}^\star),$$

where we used $\mathbf{D}^\star + \nabla F(\mathbf{X}^\star) = 0$, and the definition of $g^k$, we can write

$$\frac{1}{\alpha^k}\|\mathbf{X}^{k+1} - \mathbf{X}^*\| \geq \|\mathbf{D}^{k+1} - \mathbf{D}^\star\| - \frac{1}{\alpha^k}\|\mathbf{X}^k - \mathbf{X}^*\| - \|\nabla F(\mathbf{X}^{k+1/2}) + \mathbf{D}^\star\|$$
$$\geq \|\mathbf{D}^{k+1} - \mathbf{D}^\star\| - 2g^k\|\mathbf{D}^k - \mathbf{D}^\star\|_M \qquad (45)$$
$$\geq \frac{1}{\sqrt{\lambda_{\max}(M)}}\|\mathbf{D}^{k+1} - \mathbf{D}^\star\|_M - 2(g^k)\|\mathbf{D}^k - \mathbf{D}^\star\|_M.$$

Let us proceed lower bounding $\|\mathbf{D}^{k+1} - \mathbf{D}^\star\|_M$. Using the update of the dual variable, in the form

$$\mathbf{D}^{k+1} = \mathbf{D}^k + \frac{1}{\alpha^k}c(I - \widetilde{W})\mathbf{X}^k - c(I - \widetilde{W})(\nabla F(\mathbf{X}^{k+1/2}) + \mathbf{D}^k),$$

we obtain

$$\|\mathbf{D}^{k+1} - \mathbf{D}^\star\|_M \geq \|\mathbf{D}^k - \mathbf{D}^\star\|_M - \frac{1}{\alpha^k}\|c(I - \widetilde{W})\mathbf{X}^k\|_M - \|c(I - \widetilde{W})(\nabla F(\mathbf{X}^{k+1/2}) + \mathbf{D}^k)\|_M. \quad (46)$$

Using

$$\|c(I - \widetilde{W})\mathbf{X}^k\|_M \leq \|\mathbf{X}^k\|_{c(I - \widetilde{W})},$$

the following holds for the last two terms on the RHS of (46):

$$\frac{1}{(\alpha^k)^2}\|\mathbf{X}^k\|_{c(I - \widetilde{W})}^2 + \|c(I - \widetilde{W})(\nabla F(\mathbf{X}^{k+1/2}) + \mathbf{D}^k)\|_M^2$$
$$= (r^k)^2 \max\left(\frac{1}{(\alpha^k)^2}\|\mathbf{X}^k - \mathbf{X}^\star\|^2, \|\mathbf{D}^k - \mathbf{D}^\star\|_M^2\right)$$
$$\leq (r^k)^2 \max((g^k)^2\|\mathbf{D}^k - \mathbf{D}^\star\|_M^2, \|\mathbf{D}^k - \mathbf{D}^\star\|_M^2)$$
$$= (r^k)^2 \max((g^k)^2, 1)\|\mathbf{D}^k - \mathbf{D}^\star\|_M^2$$
$$= (r^k)^2\|\mathbf{D}^k - \mathbf{D}^\star\|_M^2,$$

where the last equality follows from $g^k \leq 1$ (as postulated in (21)).

Finally, using $\sqrt{2}\sqrt{a^2 + b^2} \geq a + b$, we deduce

$$\|\mathbf{D}^{k+1} - \mathbf{D}^\star\|_M \geq (1 - \sqrt{2}r^k)\|\mathbf{D}^k - \mathbf{D}^\star\|_M.$$

Substituting the above inequality in (45) yields the desired result. $\square$

## E.6 Proof of Lemma 11

Using (22) and (44) in (17) (while neglecting therein the last two negative terms on the RHS), yields

$$V^{k+2} \leq \left\|\mathbf{X}^{k+1} - \mathbf{X}^\star\right\|^2 + (\alpha^{k+1})^2 \|\mathbf{D}^{k+1} - \mathbf{D}^\star\|_M^2 - \mu^{k+1}\alpha^{k+1}\|\mathbf{X}^{(k+1)+1/2} - \mathbf{X}^\star\|^2$$

$$\overset{(44)}{\leq} \left\|\mathbf{X}^{k+1} - \mathbf{X}^\star\right\|^2 + (\alpha^{k+1})^2 \|\mathbf{D}^{k+1} - \mathbf{D}^\star\|_M^2 - \mu^{k+1}\alpha^{k+1}(1 - c(1 - \lambda_m(\widetilde{W})))^2\|\mathbf{X}^{k+1} - \mathbf{X}^\star\|^2$$

$$\overset{(22)}{\leq} (1 - \mu^{k+1}\alpha^{k+1}(1 - c(1 - \lambda_m(\widetilde{W}))^2)/2)\|\mathbf{X}^{k+1} - \mathbf{X}^\star\|^2 + (\alpha^{k+1})^2 \|\mathbf{D}^{k+1} - \mathbf{D}^\star\|_M^2$$

$$- \frac{\mu^{k+1}\alpha^{k+1}}{2}(1 - c(1 - \lambda_m(\widetilde{W})))^2 \frac{1}{\lambda_{\max}(M)}\left(1 - \frac{2g^k\sqrt{\lambda_{\max}(M)}}{1 - r^k\sqrt{2}}\right)^2 (\alpha^k)^2\|\mathbf{D}^{k+1} - \mathbf{D}^\star\|_M^2$$

$$\leq (1 - \mu^{k+1}\alpha^{k+1}(1 - c(1 - \lambda_m(\widetilde{W}))^2)/2)\|\mathbf{X}^{k+1} - \mathbf{X}^\star\|^2$$

$$+ \left(1 - \frac{\mu^{k+1}\alpha^{k+1}}{2(\gamma^k)^2}(1 - c(1 - \lambda_m(\widetilde{W})))^2 \frac{1}{\lambda_{\max}(M)}\left(1 - \frac{2g^k\sqrt{\lambda_{\max}(M)}}{1 - r^k\sqrt{2}}\right)^2\right)$$

$$\times (\alpha^{k+1})^2\|\mathbf{D}^{k+1} - \mathbf{D}^\star\|_M^2,$$

where the last inequality follows from $\alpha^k/\alpha^{k+1} \geq 1/\gamma^k$.

We deduce

$$V^{k+2} \leq \max\left(\frac{\alpha^{k+1}}{\alpha^k}, 1\right)^2 \left(1 - \frac{\mu^{k+1}\alpha^{k+1}(1 - c(1 - \lambda_m(\widetilde{W})))^2}{2(\gamma^k)^2 \max(\lambda_{\max}(M), 1)} \frac{1}{(\gamma^k)^2}\left(1 - \frac{2g^k\sqrt{\lambda_{\max}(M)}}{1 - r^k\sqrt{2}}\right)^2\right) V^{k+1}.$$

The final statement of the lemma follows from the above inequality and

$$V^{k+1} \leq \max\left(\frac{\alpha^k}{\alpha^{k-1}}, 1\right)^2 V^k,$$

due to (20) and $\rho_1^k < 1$. $\qquad\square$

## E.7 Proof of Lemma 12

Let us consider the case $N \geq \lceil\beta_1\rceil + 2$. Using $\gamma^k = \left(\frac{k+\beta_1}{k+1}\right)^{\beta_2}$, we can bound the product of $\gamma^k$'s as

$$\ln \prod_{k=0}^{N-1} \gamma^k = \beta_2 \sum_{k=0}^{N-1} \ln \frac{k + \beta_1}{k + 1}$$

$$= \beta_2 \sum_{k=0}^{\lceil\beta_1\rceil} \ln \frac{k + \beta_1}{k + 1} + \beta_2 \sum_{k=\lceil\beta_1\rceil+1}^{N-1} \ln\left(1 + \frac{\beta_1 - 1}{k + 1}\right)$$

$$\leq \beta_2(\lceil\beta_1\rceil + 1)\ln\beta_1 + \beta_2(\beta_1 - 1)\sum_{k=\lceil\beta_1\rceil+1}^{N-1} \frac{1}{k + 1} \tag{47}$$

$$\leq \beta_2(\lceil\beta_1\rceil + 1)\ln\beta_1 + \beta_2(\beta_1 - 1)\sum_{k=1}^{N-1} \frac{1}{k + 1}$$

$$\leq \beta_2(\lceil\beta_1\rceil + 1)\ln\beta_1 + \beta_2(\beta_1 - 1)\ln N,$$

which proves (31). Notice that the above bound holds also if $N \leq \lceil\beta_1\rceil + 2$.

Let us determine now $N_0$ such that (32) holds. Condition (32) is met if the following inequality holds

$$\ln\left(\prod_{k=0}^{N-1} \gamma^k\right)^2 + (N - 1)\ln(1 - \rho/2) \leq 0,$$

where we used that $1 - \rho \le (1 - \rho/2)^2$ for $\rho \in (0, 1)$. Bounding the LHS yields

$$\ln\left(\prod_{k=0}^{N-1} \gamma^k\right)^2 + (N-1)\ln(1-\rho/2) \overset{(47)}{\le} 2\beta_2(\lceil\beta_1\rceil + 1)\ln\beta_1 + 2\beta_2(\beta_1 - 1)\ln N + (N-1)\ln(1-\rho/2)$$

$$\le 2\beta_2(\lceil\beta_1\rceil + 1)\ln\beta_1 - \ln(1-\rho/2) + 2\beta_2(\beta_1 - 1)\ln N - \frac{\rho}{2}N$$

$$\le 2\beta_2(\lceil\beta_1\rceil + 1)\ln\beta_1 + \ln 2 + 2\beta_2(\beta_1 - 1)\ln N - \frac{\rho}{2}N.$$

It follows that (32) holds if

$$N \ge \frac{4}{\rho}(2\beta_2(\lceil\beta_1\rceil + 1)\ln\beta_1 + \ln 2)$$

and

$$N \ge \frac{8\beta_2(\beta_1 - 1)}{\rho}\ln N.$$

A sufficient condition for the last inequality to hold is

$$N \ge \frac{16\beta_2\beta_1}{\rho}\ln\frac{8\beta_2\beta_1}{\rho}.$$

This completes the proof. □

# F    Proof of the Intermediate Results in Appendix D

## F.1    Proof of Lemma 14

**1.** This assertion comes readily from the definition of $\mathcal{I}_N$ and the backtracking procedure.

**2.** Let $\mathcal{N}_i(k)$ be the set of neighbors of agent $i$ that are at most $k \ge 1$ hops away from agent $i$, including agent $i$ itself. For notational consistency, $\mathcal{N}_i(1) = \mathcal{N}_i \cup \{i\}$. Using $\mathcal{N}_i(k)$, we can rewrite the local-min consensus step of each agent $i$ at iteration $k$ as

$$\alpha_i^k = \min_{j \in \mathcal{N}_i(1)} \overline{\alpha}_j^k,$$

where $\overline{\alpha}_j^k$ is the stepsizes produced by the line-search of agent $j$ at iteration $k$.

Let $\overline{k} > k$ be an iteration such that $k + 1, \ldots, \overline{k} \notin \mathcal{I}_N$. It must be

$$\alpha_{\min}^t = \gamma^t \alpha_{\min}^{t-1}, \quad \forall t = k + 1, \ldots, \overline{k}.$$

In particular, this implies

$$\overline{\alpha}_i^t = \gamma^t \min_{j \in \mathcal{N}_i(1)} \overline{\alpha}_j^{t-1}, \quad \forall t = k + 1, \ldots, \overline{k},$$

which leads to

$$\alpha_i^{\overline{k}} = \left(\prod_{l=k+1}^{\overline{k}} \gamma^l\right) \min_{j \in N_i(1 + \overline{k} - k)} \overline{\alpha}_j^k.$$

Finally, taking $\overline{k} = k + d_{\mathcal{G}}$ and noting $N_i(d_{\mathcal{G}}) = [m]$, we proved the second statement of the lemma.

**3.** From the backtracking line-search and the definition of $\mathcal{I}_N$ it follows that

$$\alpha_{\min}^k \begin{cases} = \gamma^k \alpha_{\min}^{k-1}, & \text{if } k \notin \mathcal{I}_N; \\ \le \frac{\gamma^k}{2}\alpha_{\min}^{k-1}, & \text{if } k \in \mathcal{I}_N. \end{cases}$$

Applying the above relation iteratively, yields

$$\alpha_{\min}^N \le \alpha^0 2^{-|\mathcal{I}_N|} \prod_{k=0}^{N-1} \gamma^k.$$

At the same time, it follows from Lemma 3 that

$$\alpha_{\min}^N \geq \min\left(\frac{\delta}{2L}, \gamma^k \alpha^0\right).$$

Combining the lower and upper bounds above, yields the desired result

$$|\mathcal{I}_N| \leq \max\left(\ln \alpha_0 L + \ln \prod_{k=0}^{N-1} \gamma^k + \ln \frac{2}{\delta}, 0\right).$$

$\square$

### F.2 Proof of Lemma 15

From (34), it follows

$$\left\lfloor \frac{N}{N_\varepsilon + d_{\mathcal{G}}} \right\rfloor > |\mathcal{I}_N|.$$

Then, according to the Dirichlet's principle there exists two iteration indices $k_1$ and $k_2$ such that

1. $\forall k \in [k_1, k_2] \Rightarrow k \notin \mathcal{I}_N$; and
2. $k_2 - k_1 \geq N_\varepsilon + d_{\mathcal{G}}$.

Invoking Lemma 14.(1) and Lemma 14.(2), it follows that all agents' stepsizes reach consensus after $k_1 + d_{\mathcal{G}}$ iterations and remain consensual for the subsequent $N_\varepsilon$ iterations. One can then invoke Lemma 13, and conclude

$$\left\|\mathbf{X}^{k_2} - \mathbf{X}^\star\right\|^2 + \frac{1}{4L^2}\|\mathbf{D}^{k_2} - \mathbf{D}^\star\|_M^2 \leq \varepsilon.$$

This concludes the proof. $\square$

## G  Additional Numerical Results

This section presents additional experiments for the Ridge Regression problem, introduced in Section 6. Here, we consider additional graph topologies, namely: 1) Ring Graphs; 2) Random Regular Graphs with degree 3; and 3) Random Regular Graphs with degree 10. The rest of the setup (including algorithms' tuning) is the same of that described in Section 6.

The experiments are summarized in Fig. 3. The findings corroborate the conclusions presented in

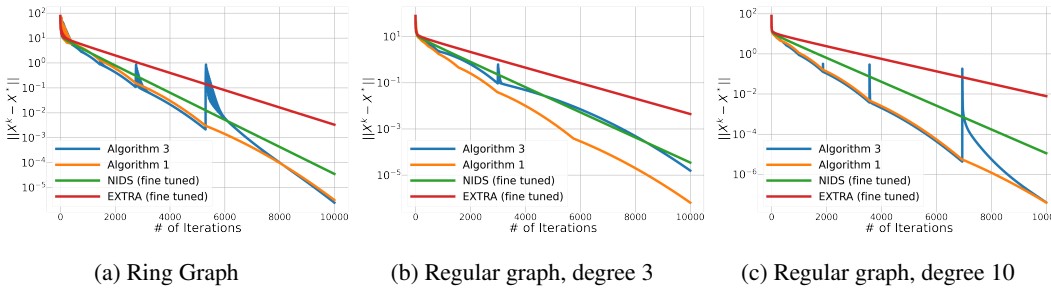

(a) Ring Graph          (b) Regular graph, degree 3          (c) Regular graph, degree 10

Figure 3: **Ridge regression** on different **regular graphs**: (3a) Ring graph; (3b) Random Regular Graph, with degree 3; (3c) Random Regular Graph with degree 10.

Sec. 6: both Algorithm 1 and Algorithm 3 outperform EXTRA and NIDS, which were finely tuned for rapid practical convergence. Quite interestingly, the performance of the proposed methods appears to be less affected by network topology and depends primarily on network connectivity.

