# OpenReview forum: "Achieving Linear Convergence with Parameter-Free Algorithms in Decentralized Optimization"
_NeurIPS.cc/2024/Conference — NeurIPS 2024 poster_

### Official Review · Reviewer_E8C7 · 2024-06-23

**Soundness:** 3
**Presentation:** 2
**Contribution:** 3
**Rating:** 5
**Confidence:** 2

**Summary:**

The paper introduced a new parameter-free algorithm based on forward-backward splitting technique and variable metric for decentralized learning problems for convex locally smooth functions. Convergence guarantee with favorable rate and analysis are provided.

**Strengths:**

1. The paper proposed the first parameter-free decentralized training algorithm integrating line search and splitting technique.
2. Convergence guarantee and analysis under milder conditions than previous studies are provided.
3. The guidelines of design seems to be of independent interest and further investigation.

**Weaknesses:**

1. The proposed algorithm is complicated, and it seems that in each iteration, a lot of computation needs to be carried out. How does the algorithm compares to the previous ones in terms of computational complexity?
2. The proposed algorithm needs more experimental evaluation.
3. In certain parts, the paper is a bit hard to follow.

**Questions:**

See weaknesses.

**Limitations:**

No.

---

> ### Author Rebuttal · Authors · 2024-08-02
>
> We thank the Referee for reviewing our work and the positive assessment on the novelty of the paper. Our reply to her/his comments/questions follows.
>
> 1.  **"The proposed algorithm is complicated":** The proposed algorithm has comparable communication cost  (step S.1 and S.2) and computational complexity (gradient evaluation) of all  existing notorious decentralized algorithms that converge to an *exact* solution of the optimization problem, such as  EXTRA, NIDS, and Gradient Tracking, *except for the line-search procedure*, which results in some extra computation in the evaluation of the function value. This extra computation is what allows one to achieve adaptivity of the algorithm, that is, convergence with **no knowledge** of any optimization or network parameter. On the contrary, **all** the existing decentralized algorithms require  **full  knowledge** of the optimization and network parameters to converge. In practice, this information is not available at the agents' side, and needs to be acquired, if one wants to implement these schemes. This calls for some nontrivial procedure that produces reasonable estimates of these optimization and network parameters,  which results in additional computation and communication costs.
> On our side, the backtracking procedure is the only extra (reasonable) computational cost one needs to pay, to obtain for the first time a decentralized algorithm that does not require any centralized knowledge and implementable in practice without any additional procedure.  In the future, it will be interesting to replace the line-search with some other adaptive procedures that require less function evaluations.
>
> 2. **"The proposed algorithm needs more experimental evaluation"**: We thank the Referee for her comment. We will expand the numerical evaluation, along the following directions:  i) logistic and ridge regression problems on other network topologies; ii) a new comparison between the proposed method and EXTRA and NIDS where for the latter we hand-tune the stepsize for the best practical performance (as requested by other Referees); and iii) simulation (and comparison with the aforementioned schemes) of a newly added adaptive method wherein the global min consensus is now replaced with the local one (to address some concerns of Referee H817).
> We added in the general rebuttal section a pdf with some of the above results. We kindly refer the Referee to that section for more details on the experiments and the figures.  We hope that the additional experiment will address the Referee's request. We are open to suggestions, if the Referee had something else in mind for the experimental evaluation.
>
> 3. **"In certain parts, the paper is a bit hard to follow":** We will be happy to improve the presentation and provide all the necessary clarifications, if the Referee can be more specific on which part is ``hard to follow''. We appreciate if the Referee can point us to the parts that are not clear to her/him.
>
>  4. We would like to stress one more time that this is the first attempt to bring adaptivity in the decentralized setting providing a systematic approach, and algorithms provably achieving linear convergence.   The main challenge in bringing adaptivity in the  *decentralized* setting is to identify in (existing) distributed algorithms a (descent) 'direction' (and local merit function) along with performing the adaptive step-size selection. The tuning of the stepsize in decentralized algorithms must depend on network properties, as stepsize values similar to centralized settings generally lead to divergence. Thus, any candidate direction should contain information on the optimization *and* network. There is no understanding of how the network should influence the direction and how to encode in the candidate direction optimization and network information. We offer the first principle-based procedure to resolve these issues, addressing the major challenge that has prevented the migration of centralized adaptive stepsize techniques to the decentralized setting. Hopefully this will be trigger new works in this direction.  Last but not least, our algorithm is provable convergent also when applied to functions that are **only locally** smooth (and locally strongly convex), which is not the case for the majority of existing decentralized algorithms, e.g., EXTRA and NIDS. This enlarges significantly the classes of problems our scheme can be applied.
>
> We would appreciate an assessment of the paper from the Referee based on this important contribution.
>
> Thanks again for the feedback on the paper. Please let us know if our answer and additional posted material satisfactorily have addressed all the Referee's comments/requests.

---

> > ### Comment · Reviewer_E8C7 · 2024-08-10
> > **Thank you**
> >
> > Thanks for the response! I have read the rebuttal and decided to maintain my score.

---

### Official Review · Reviewer_ioM9 · 2024-07-10

**Soundness:** 2
**Presentation:** 1
**Contribution:** 2
**Rating:** 5
**Confidence:** 3

**Summary:**

This paper proposed a parameter-free method for decentralized learning and showed that the method converges to the optimal solution linearly without hyperparameter tuning.

**Strengths:**

This paper proposed a novel decentralized method and the convergence rate is analyzed under the general setting.

**Weaknesses:**

1. Theorem 4 provided the convergence rate of the proposed method without hyperparameter tuning, but Corollary 4.1 provides the results with some $\beta_1$ and $\beta_2$. Thus, the reviewer thinks that the results in Corollary 4.1 require hyperparameter tuning. The author discussed the convergence rate in Corollary 4.1, but this result is not the convergence rate of the parameter-free method. The reviewer thinks that it is necessary to discuss the results in Theorem 4 and compare these results with one of the existing methods.
2. The reviewer does not understand what the author would like to conclude from Theorem 5 (in the weakly convex setting). Theorem 5 only shows that the amount of parameter updates (i.e., $\| X^j - X^{j+1} \|^2$) is bounded from above by some factor and does not show that the proposed method converges to the optimal solution.
3. The author did not tune the stepsize for EXTRA and NIDS in Fig. 1 and did not mention how to tune the stepsize in Fig. 2. The author claimed that the proposed method consistently outperforms EXTRA and NIDS in Figs 1 and 2, but these results are not convincing because the results of NIDS and EXTRA are suboptimal, at least in Fig. 1. The reviewer would like to see the results of EXTRA and NIDS with well-tuned stepsize and see the comparison between the proposed method and the prior methods.
4. The line 3 in Algorithm 3 can not be calculated in a decentralized manner.

**Questions:**

See the above comments.

**Limitations:**

See the above comments.

---

> ### Author Rebuttal · Authors · 2024-08-02
>
> We thank the Referee for her/his comments, which will help us to clarify some parts of the paper  as well as improve the revised version. Our detailed reply follows.
>
> 1. **The reviewer thinks that the results in Corollary 4.1 require hyperparameter tuning"**: We apologize for this misunderstanding, due to the lack of adequate comments from our side. **There is no need of  hyperparameter tuning here**. The two parameters $\beta_1\geq1$ and $\beta_2>0$ are just *extra* degrees of freedom to offer further flexibility in the algorithm design (see comment below), but they are not necessary. In fact, **(i)** the theoretical convergence rate as in Th. 4 (or Corollary 4.1) does not depend (asymptotically) on $\beta_1$ and $\beta_2$. **(ii)** One can set $\beta_1=\beta_2=1$ and they will disappear from the algorithm. We could have introduced the algorithm directly setting   $\beta_1=\beta_2=1$.  Notice that all our simulations are conducted under this choice, $\beta_1=\beta_2=1$, which provides quite compelling performance.
> **Further comments on  $\beta_1$ and $\beta_2$**: at the high level, the introduction of these two parameters is to allow practitioners to use 'larger' stepsizes $\alpha_i^t$   out of the line-search, via the   *nonmonotone* sequence $\{\gamma^k\}$, where $\gamma^k=\left(\frac{k+\beta_1}{k+1}\right)^{\beta_2}$. This helps in practice to achieve better performance, if one is willing to explore this extra degree of freedom. This is however not necessary for the theoretical convergence and, as mentioned, one can either choose $\beta_1=\beta_2=1$ or directly set $\gamma^k=1$ for all iterations $k$. We discussed this matter in lines 207-211 of the original submission. We will further clarify this aspect  in the revised version of the paper.
> In general we talk about `` hyperparameter tuning''  when the parameter involved are not free but must satisfy some conditions *coupling* them with other parameters of the algorithm  (such as the stepsize, etc.). In that case, their choice would be no longer free.  This is not the case here, because there are no such requirements on $\beta_1$ and $\beta_2$, but $\beta_1\geq1$ and $\beta_2>0$ (hence not restricting the choice of any other algorithm tuning parameter).
>
> 2. **About  Theorem 5 (in the weakly convex setting)**: We thank the Referee for the comment. The result establishes the asymptotic optimality of any limit point of the sequence generated by the algorithm. However, we agree that the nonasymptotic convergence result is not particularly strong. Given that the strongly convex case is more challenging in terms of guaranteeing linear convergence, and considering the new anticipated results from other Referees' comments, we will remove the convex case. This will allow us to use the space to introduce the local min-consensus variant and its convergence, and possibly the stochastic case under the Retrospective Approximation framework. We hope that the Referee will agree that the convex case is not the major result of the paper.
>
> 3. **The author did not tune the stepsize for EXTRA and NIDS in Fig. 1 ...:** It is challenging to fairly compare our scheme with existing ones because the existing decentralized schemes require full knowledge of the network and optimization parameters, whereas our scheme does not.  We agree that other choices could have been made for  the tuning of EXTRA and NIDS.  Some question to guide the process are:
> **(i)** Should the comparison be done using the *theoretical* tuning recommended in the papers to ensure convergence? By doing so, the comparison would be fair in the sense that all algorithms are guaranteed to converge. However, this approach may result in quite stringent stepsize values
> Or **(ii)** Should one take a more practical approach, ignoring theoretical conditions (and thus convergence guarantees) and hand-tuning for the best observed convergence?
>
> Both approaches introduce some level of unfairness. In our paper, we followed the former approach: we used the tuning recommended in the respective papers for our scheme, EXTRA, and NIDS to guarantee convergence. However, it can still be argued that the comparison is unfair because NIDS and EXTRA require full knowledge of the network and optimization parameters, while our scheme does not. But in this case, the unfairness is towards our scheme.
>
> To address the Referee's suggestion, we have conducted new experiments where we hand-tuned EXTRA and NIDS for their best practical performance and compared them with our method. Note that in this setting, EXTRA and NIDS lack theoretical convergence guarantees, while our method maintains them. We kindly refer the Referee to the global rebuttal section for more details on the experiments and the new figures (attached pdf therein). We are open to conduct alternative comparisons if the Referee has something different in mind to share.
>
> 4.**The line 3 in Algorithm 3 can not be calculated in a decentralized manner.** This method is actually implementable in the existing wide area networks, thanks to the LoRa technology. This has been briefly discussed in the paper  (lines 212-225) and further elaborated in the Reply to Reviewer H817, please refer therein to item 1 **About the global min-consensus**. Also, we provided a variant of the algorithm that replaces the global min-consensus with a **local** min-consensus (see **From the global min-consensus to the local min-consensus** in the reply to Reviewer H817). This variant has been simulated and results provided in the attached pdf in the section of general comments.
>
> Hope the Referee is willing to reconsider her/his initial assessment, cosidering that this is the *first* adaptive methods in the decentralized literature, even just focusing on strongly convex problems. Achieving adaptivity posed several challenges, resolved in this work for the first time. We elaborated further on this aspect in point 3) of our reply to Reviewer H817, which we refer the Referee to.

---

> ### Comment · Reviewer_ioM9 · 2024-08-10
>
> We thank the authors for their response.
> All concerns were addressed after reading the authors' replies.
>
> The reviewer thinks the proposed methods converge without any hyperparameter-tuning, which is a strong advantage over EXTRA and NIDS. However, it is reasonable that the parameter-free methods are inferior to these methods with well-tuned hyperparameters, and the reviewer guesses that many readers would like to see how large this gap is. Thus, it is important to discuss this gap in the experimental section.
>
> All reviewer's concerns were solved. The reviewer raised the score to 5.

---

> > ### Author Response · Authors · 2024-08-10
> >
> > Thanks for reading the rebuttal and reassessing the evaluation of the paper. We will expand the numerical section and properly comment the comparison, as requested.
> > As followup of your last comment, we wish to remark that it is not obvious that a nonadaptive, even grid tuned, decentralised algorithm is superior to an adaptive one. The reason is that, even if grid-search tuned, the step size in a non adaptive algorithm is chosen ‘once and for ever’. On the contrary, in our adaptive algorithm, the step size changes at each iteration. This offers the possibility to use larger step size vales when traveling over part of the landscape with favourable Lipschitz gradient constant. This is not the case with fixed (albeit tuned) step size algorithm, which in general are forced to use much  smaller step size.  We think that it will not be difficult to construct case-study functions enhancing this difference.

---

### Official Review · Reviewer_H817 · 2024-07-11

**Soundness:** 1
**Presentation:** 2
**Contribution:** 1
**Rating:** 6
**Confidence:** 4

**Summary:**

This paper studies adaptive parameter determination in decentralized optimization. It is a meaningful and interesting topic to investigate. They propose a decentralized method to solve consensus optimization, develop an adaptive parameter strategy, and show the linear convergence of their algorithm.

**Strengths:**

The main strength of the paper lies in the high value of the topic they studied, which is indeed a very meaningful and also a difficult topic to explore.

**Weaknesses:**

I was very excited when reading the abstract of this paper, because I know the difficulty of this topic. I was worried about the global min-consensus step, while the authors developed a local min-consensus strategy to replace it, which well addressed my concerns.

**Questions:**

No

---

> ### Author Rebuttal · Authors · 2024-08-02
>
> We thank the Referee for the review. We kindly disagree with her/his assessment, which is an oversimplification and trivialization of our contributions, missing the challenges our work addresses.  Details follow.
>
> 1)**About the global min-consensus:** The Referee's sole concern is the presence of a min-consensus step in the algorithm, whereby agents evaluate the minimum of their local stepsizes over the network at each iteration. The Referee questions the implementability of this step in practice. **The reality is quite the opposite**. As discussed in our paper (lines 212-225), the min-consensus step **is implemented seamlessly** in commercial wide-area mesh networks without changes to existing communication protocols or additional hardware. This is facilitated by the  *decade-old* Long Range (LoRa), commercialized by Semtech and widely integrated into most commercial transceivers in wide-area networks. The Referee may want to know that LoRa is used in various fields, including industry, smart cities, environmental monitoring, agriculture, healthcare, and long-range, low-power Internet of Things (IoT) applications [2, 14, 15] (references as in our paper). LoRa supports communication ranges of hundreds of kilometers in free space, hundreds of meters in indoor environments, and half a kilometer in urban settings [14], with a maximum data rate of hundreds of kbit/s on average and a few kbits/sec at the longest possible range. **This technology is ideal for implementing the min-consensus step**: each agent broadcasts a few bits representing the quantization of its stepsize over the LoRa channel, which reaches all other agents in the network thanks to the extensive range of the LoRa signal. Each agent can then compute the minimum of all the stepsizes. **This resolves the issue of implementability raised by the Referee**. This is not merely our claim; it is a fact backed by existing technology. The Referee cannot overlook this well-established technology and solution.
>
> 2)**From the global to the *local* min-consensus:** Intellectually, we agree that developing a method without the global min-consensus is worthwhile. We can address this by providing a first variant of the algorithm where the *global* min-consensus is replaced by a **local** min-consensus. Specifically, we replace  step 3 of Algorithm 1 with
> $\\alpha^k_i=\\text{min}_{j\\in\\mathcal{N}_i}$, where $\\mathcal{N}_i$ is the the set of neighbors of agent $i$ (including agent $i$ itself).  Then, in the other steps the common stepsize is replaced by a diagonal matrices containing the individual stepsize $\alpha^k_i$ above.
>
>
> The convergence analysis can be readily adapted to this variant (see box 'Official comments' below), yielding: the number of iterations $N$ for  $\\min\_{k\\in \\{1,...,N\\}} V^k\\leq \varepsilon$ reads $$N=\\widetilde{\\mathcal{O}}(d\_{\\mathcal{G}}N\_{\\varepsilon}),$$
> where $N_\\varepsilon$ is the number of iterations achieved using the *global* min-consensus (this is what we defined as $N$  in Corollary 4.1, for the three cases) and $d_{\\mathcal{G}}$  is  the diameter  of the graph. This shows that if a global consensus is replaced by a min-consensus, there is a  degradation of the overall number of iterations by a  factor $d_\\mathcal{G}$.  This is not surprising, because it is known that  a min-consensus algorithm convergences on a network in a *finite* number of iteration of the order of the diameter. We believe that the multiplicative dependence of the convergence rate on $d_\\mathcal{G}$ can be improved or removed, see the box 'Official comments' below.
>
> We tested this new variant running new simulations reported in the attached pdf in the global comment section.
>
> We hope that this satisfies the intellectual curiosity of the Referee. We can add this variant to the revised paper, if the Referee wishes so.
>
> 3)**The Referee is overlooking our major contributions:** Our contributions cannot be trivialized to the min-consensus step only. In fact, equipping existing decentralized algorithms such as DGD, EXTRA, or NIDS with a global min-consensus procedure *does not grant them adaptivity*; even worse, it will jeopardize their convergence. **It is not even clear how to identify for them a   (descent) 'direction'**  (and local merit function)  along with  performing the adaptive step-size selection,  see lines 190-195 in the paper. The  tuning of the stepsize in decentralized algorithms must depend on network properties, as stepsize values similar to centralized settings generally lead to divergence. Thus, any candidate direction should contain information on the optimization **and** network. There is no understanding of how the network should influence the direction and how to encode in the candidate direction optimization and network information. **We offer the first principle-based procedure to resolve these issues, addressing the major challenge that has prevented the migration of centralized adaptive stepsize techniques to the decentralized setting**. Hopefully this will be trigger new works in this direction.
> Last but not least, our algorithm is provable convergent also when applied to functions that are **only locally smooth** (and locally strongly convex), which is not the case for the majority of existing decentralized algorithms, e.g., EXTRA and NIDS. This enlarges significantly the classes of problems our scheme can be applied.
>
> We hope this clarification helps the Referee appreciate the importance of our contribution.
>
> **In summary:**  **(i)**  We proved that *existing technology supports global min-consensus for free*. **(ii)** Still, we addressed the Referee's comment by providing and numerically testing a variant of the algorithm that removes global min-consensus. **(iii)** We clarified that our major contribution is not the global min-consensus and demonstrated that even with a global consensus step, existing schemes do not achieve adaptivity for free,  let alone convergence guarantees.

---

> > ### Comment · Reviewer_H817 · 2024-08-08
> > **Thanks for your strong rebuttal. I decided to raise my score**
> >
> > Thank you for your efforts in responding my comments. This is a strong rebuttal and the "local min-consensus" addressed my concerns. Therefore, I decided to raise my score.
> >
> > However, I insist that the global min-consensus step is not preferable even if, as the authors discussed, that it has been applied in many real applications. This global consensus step requires multiple rounds of local min-consensus steps, which is similar to inner iterations inside each global outer iteration. Each inner iteration requires synchronization of all nodes, while synchronization in networks is not easy and do have costs, especially when the network size is large.
> >
> > Except the above, I do not have concerns, but have one question. Could the authors explain why the optimality errors of using the local min-consensus strategy oscillate seriously in your experiments?

---

> > > ### Author Response · Authors · 2024-08-08
> > > **Thank you**
> > >
> > > We wish to thank the Referee for spending extra time to go over our rebuttal and reassessing the evaluation of our work. We are glad that the new suggested local min-consensus addresses his/her concerns. Thank you! Of course, we will add the local min-consensus and new simulations to the paper. Likely, we will remove the treatment of the convex case for the sake of space.
> > >
> > > Below we address the new questions/comments.
> > >
> > > -**spikes in the error dynamics over the iterations:** The reasons why numerically we observe the `spikes' in the error dynamics is because the local min-consensus is a  'static' procedures that estimate the global min among the stepsizes but cannot track its variations smoothly. We conjecture that we can cope with this issue putting forth *dynamic* estimation mechanisms of the global min-stepsize, which are capable to track time-varying signals. We will definitely investigate this topic in the near future, aiming at comparing alternative strategies to track time-varying global-min consensus over networks. We thank one more time the Referee for triggering this interesting direction.
> > >
> > > -**Few more comments on the implementability of the global min-consensus:** Based upon the new comments from the Referee on this aspect, we feel that some further clarifications may help. When we mentioned that LoRa allows one to readily implement global-min consensus 'instantaneously' over a mesh network, we forgot to point that  it generates a *fully connected* network (wherein each node can reach everybody else) of low-throughput channels that **co-exists** simultaneously and continuously with the high-throughput mesh network whereby agents can only communicate with their immediate neighbors. This means that (i) the global min-consensus  is reached in  *one* iteration (everybody transmits its own stepsize and everybody receives the stepsize of all the others); this is because the LoRa network is a fully-connected graph (ii) The two type of communications, the stepsize broadcasting over the LoRa to any other agent and the transmission of the vector variables to each immediate neighbor, happen at the same time (not sequentially) because using different channels and protocols.  The Referee may think of this system like two overlapping. co-exiting graphs, one complete (over which the stepsize are exchanged) and one not, modeling the mesh networks wherein vector variables are communicated among neighbors. However we agree with the Referee that some synchronism in the implementation of the algorithm is necessary, although there are no (conceptual)  double-loops.
> > >
> > > In the revised version, we will rewrite the paragraph discussing this matter as well.
> > >
> > > Thanks again.

---

> ### Author Response · Authors · 2024-08-02
> **Some technical details on the discussed new local-min consensus procedure**
>
> For the sake of completeness, we show  how to modify the existing proof in the paper, to account for the replacement of the global min-consensus with the local min-consensus. The changes are minor. Below we will use the same notation and equation numbering as in the paper.
>
> **Step 1:** Using the same merit function $V^k$ as in the paper (see eq. (11)) and following the steps of the proof of Th. 4, one can easily check that the decay of $V^k$  now becomes  $$V^{k+1}\leq (1-\rho^k)(\gamma^k)^2 V^k + C (\\alpha_{\\max}^k-\\alpha_{\\min}^k) V^k+ (\\alpha_{\\max}^k-\\alpha_{\\min}^k) R^k \\qquad (A.1),$$
> where $\\alpha_{\\max}^k$ (resp $\\alpha_{\\min}^k$) is the largest (resp. smallest) stepsize among the agents at iteration $k$,  $C$ is a constant independent on the iteration index, and $R^k$ depends on $X^k$ and $D^k$ and is uniformly bound when $X^k$ and $D^k$ are so; let then $R$ such that $R\geq R^k$, for all $k$. Comparing (A.1) above with (12) in the paper, we notice that now in the decay of  $V^k$ there is a second and third  term on the RHS of (A.1), both due to the use of local min-consensus that no longer guarantees that all the stepsizes are equal at each iteration. The rest of the analysis consists in studying the dynamics of this extra error terms.
>
> **Step 2:** Based upon $\\alpha_{\\max}^k-\\alpha_{\\min}^k$ in (A.1), we naturally  identify at each iteration $k$, a favorable and an  undesired event, namely:
>
> 1. **Favorable event:** $\\alpha_{\\max}^k-\\alpha_{\\min}^k=0$. Substituting in (A.1), yields $V^{k+1}\\leq (1-\\rho^k/2)(\\gamma^k)^2 V^k$, which is  the same dynamic of (12) in the paper, achieved under global min-consensus. This means that, subject to this event, the algorithm behaves like performing a global min consensus.
>
> 2. **Unfavorable event:**  $\\alpha_{\\max}^k-\\alpha_{\\min}^k> 0$.
>
> The following facts hold for these two events:
>
>
> $\\bullet$ **Fact 1 (favorable event):** The proof in the paper shows that a number of  $N\_{\\varepsilon}$ *consecutive*  iterations  of the favorable event are sufficient to guarantee that $V^k\leq \varepsilon$, where $N\_{\\varepsilon}$ is given by $N$ as defined as in Corollary 4.1 in the paper (for all the three cases there).
>
> $\\bullet$ **Fact 2 (unfavorable event):**  Invoking the finite-time convergence of the min-consensus algorithm, it is not difficult to check that the unfavorable event can happen at most  $2 d_{\\mathcal{G}}(\\log k+\log(L/L^0))$ times in $k$ iterations.
>
> **Step 3:** Combining  Fact 1 and Fact 2, we can conclude that if the number of total iteration $N$ is large enough, namely $$\\frac{N}{2d\_{d\_{\mathcal{G}}}\\left(\\log N + \log (L/L^0)\\right)}>N_{\varepsilon},\\quad (A.2)$$ then it must hold $$\\min\_{k\\in \\{1,\\ldots, N\\}} V^k\\leq \\varepsilon.$$
>
> The following is sufficient for (A.2) to hold:  $$N=\\mathcal{O}\\left(\\max\\left\\{\\log d\_{\\mathcal{G}}+ \\log N\_\\varepsilon, \\log L/L^0\\right\\}d\_{\\mathcal{G}} N\_{\\varepsilon}\\right)=\\widetilde{\\mathcal{O}}(d\_{\\mathcal{G}}N\_{\\varepsilon}),$$ where $\\widetilde{\\mathcal{O}}$ neglects log-factors (independent on $\\varepsilon$). This completes the proof.
>
> We believe that the linear dependence of the number of iterations on $d_\\mathcal{G}$ can be improved to $\log d_\\mathcal{G}$, providing a finer analysis of the frequency of the undesired event (Fact 2). Furthermore, we conjecture that the dependence on  $d_\\mathcal{G}$ can be eliminated (as multiplicative effect of $\\log 1/\\varepsilon$) if instead of using **static** procedures to estimate the global min among the stepsizes, like the local min-consensus, *dynamic* estimation mechanisms of the global min-stepsize are put forth. This is the subject of future investigations, and beyond the scope of this paper. We provided this result and proof only to satisfy the curiosity of the Referee.

---

> > ### Comment · Reviewer_H817 · 2024-08-08
> >
> > Thank you for the detailed explanation. I do wish to see the new local min-consensus strategy in the final manuscript.

---

### Official Review · Reviewer_aLXV · 2024-07-12

**Soundness:** 4
**Presentation:** 3
**Contribution:** 3
**Rating:** 7
**Confidence:** 4

**Summary:**

This paper introduces a new algorithm for decentralized optimization.  The main advantage over previous work in this domain is that it allows adaptive stepsize selection (via backtracking) that is independent of the properties of the functions being minimized.  The analysis of the algorithm recovers the linear rate for convergence for functions that are smooth and strongly convex, and the standard sublinear rate for the convex case.

**Strengths:**

As the authors state, this seems to be the first fully decentralized algorithm with adaptive stepsize selection that has strong convergence guarantees.  I found the operator splitting method for solving the KKT conditions innovative, and it is nice that this leads naturally to a way to do local stepsize adaptation.

**Weaknesses:**

While the numerical experiments are sufficient, new results on ridge and logistic regression can only get one so excited.

The exposition of the saddle point reformulation at the beginning of Section 3 could be improved.
Line 142: Does the optimization in (P') involve y at this point?  Should x be capitalized?
Line 143: Some discussion about why K is being introduced here would help the exposition a lot, along with a little more on why the solutions of (P) and (P') match.  A lot of work is being left to the reader at this critical point.

**Questions:**

It is outside the scope of this submission, but I would be interested to know if this type of adaptive stepsize selection made a difference in practice in the stochastic setting.

**Limitations:**

None.

---

> ### Author Rebuttal · Authors · 2024-08-02
>
> We thank the Referee for reviewing our paper and her/his positive assessment. We are glad that she/he recognized the major novelty of the proposed approach, i.e., the novel operator splitting technique that naturally leads to local stepsize adaptation.
>
> Our reply to her/his questions follows.
>
> 1. **Simulations:** We agree that more simulations will be useful. We plan to add new experiments as follows: **i)** logistic and ridge regression problems on other topologies; **ii)** a new comparison between the proposed method and EXTRA and NIDS where for the latter we hand tune the stepsize for the best practical performance (as requested by other Referees); and **iii)** simulation (and comparison with the aforementioned schemes) of a newly added adaptive method wherein the global min consensus is now replaced with the local one (to address some concerns of Referee H817).
>
> 2. **Exposition:** We thank the Referee for her/his feedback. **i)** The Reviewer is right, in Line 142, there is a typo: y should not be there and x should be capitalized. **ii)** We will provide some insight on why K is introduced in (P'). This goes along with some clarification on why (P) and (P') have the same solution, under the condition that K satisfies condition (c2) (which serves exactly this purpose). We will clarify these aspects in the revised manuscript.
>
>
>
> 3. **Adaptivity in the stochastic setting:** The difficulty in implementing adaptivity in the stochastic setting lies in the line-search procedure, which is notoriously difficult in the presence of noise. This is a subject of our future investigation.
> However, a positive answer to the Reviewer's question can be  provided for the class of stochastic optimization problem for which a *Retrospective Approximation Approach* is feasible for the line search.  Specifically, at each iteration $t$, a sample approximation $f_i(x_i; \xi_i^t)$  and $\nabla f_i(x_i; \xi_i^t)$ is constructed for the population function $f_i(x_i)=\mathbb{E}_{\xi}[f_i(x;\xi)]$ and its gradient $\nabla f_i(x_i)$ ($\xi_i^t$ is the sample batch drawn by agent $i$ at iteration $t$), and the proposed  local line-search procedure is performed using  now $f_i(\bullet; \xi_i^t)$  and $\nabla f_i(\bullet; \xi_i^t)$. The difference with a classical stochastic line-search is that here the batch sample $\xi_i^t$ is kept *fixed* during the entire backtracking procedure (of course it can change from one iteration to another).  Within this setting, all the results of the paper are preserved if read in expectation.  While this Retrospective Approximation Approach does not cover stochastic oracles in their generality, it is a reasonable model for certain machine learning problems, for example those where evaluating $f_i(\bullet; \xi_i^t)$  and  $\nabla f_i(\bullet; \xi_i^t)$ at a given point $x$ corresponds to perform one pass on the data batch $\xi_i^t$.
> Quite interestingly, in an early version of the manuscript, we had this result, which was removed in the final version because of space limit. We can add it back, maybe removing the study of deterministic convex functions.
>
> We thank again the Referee for her/his valuable comments.

---

> > ### Comment · Reviewer_aLXV · 2024-08-13
> >
> > Thanks for this, especially your insights into the stochastic setting.

---

### Author Rebuttal · Authors · 2024-08-06

We thanks all the Referees for reviewing our paper and their feedback and suggestions. We did our best to address any concern in the individual reply, which we refer to for details. Below we only discuss the new added experiments.

A common request from multiple Referees has been expanding the experiments, which we have done it. Some new experiments  (more in the revised paper) are reported in the attach a pdf.  More specifically, we did the following:

1. As requested by one Referee, we have changed the tuning of the benchmark (nonadaptive)  algorithms, NIDS and EXTRA. We now select the stepsize by **hand-tuning** aiming for the best practical performance (fastest observed rate). Notice that in this case, NIDS and EXTRA are *no longer* guaranteed to convergence theoretically, because stepsize values typically violates the conditions for convergence. On the contrary our scheme has theoretical convergence guarantees. Furthermore, and quite remarkably, it compares favorably even with this hand-tuned instances of NIDS and EXTRA.

2. We have introduced and  simulated also a new version of our algorithm, that replaces the global min-consensus with the **local** min consensus. This is to address a comment from the Reviewer H817 who considers the global min-consensus not desirable (we kindly disagree with this assessment, as *current technology supports scalar min-consensus over network for free*). The new variant still has good practical performance and convergence guarantees.

3. We have compared  the above decentralized algorithms (NIDS, EXTRA, our original scheme based on global min-consensus, and the new variant based on local min-consensus) over different graph topology (not present in the original submission), namely (a) a line graph, (ii) a ring, and (iii) a random regular graph  (with two degree values).  On each graph we report experiments for logistic and ridge regression problems.

**Some comments on the numerical results**

1. The figures demonstrate that our original scheme, based on global min-consensus, consistently outperforms EXTRA and is almost always superior to NIDS, even when these algorithms are hand-tuned. This holds true across different optimization problems and network topologies. This is particularly noteworthy considering that hand-tuning is neither desirable nor easily implementable in practice, especially when the network topology is unknown. Additionally, hand-tuning generally compromises theoretical convergence guarantees. On the contrary our schemes do not require any intervention and more importantly have theoretical guarantees.

2. The new variant of the proposed adaptive algorithm based on local min-consensus still shows strong performance, holding well against hand-tuned EXTRA and NIDS. **(i)** While the convergence rate remains linear, the error curves may exhibit some 'spikes.' This is consistent with the new convergence analysis provided (see details in the Reply to Reviewer H817) and is due to the min-consensus converging in a finite number of iterations proportional to the graph diameter. The correct way to interpret the convergence is that the minimum of the error gap within N iterations falls below the desired accuracy *geometrically fast*. **(ii)** As expected, as the graph becomes more connected, the new variant of the algorithm using local min-consensus matches the performance of the original algorithm using global min-consensus.

3. On convex problems (logistic regression) the gap in performance of the different algorithms (but EXTRA) is less pronounced than that  in the strongly convex case.

---

### Decision · Program_Chairs · 2024-09-25

**Decision:**

Accept (poster)

**Comment:**

This work studies decentralized optimization and proposes a method that utilizes backtracking line search to avoid tuning the step size parameters. Unlike previous work, this algorithm's convergence does not require step sizes to be dependent on problem parameters, making it adaptive. A linear convergence rate is established for strongly convex costs, and sublinear rates are demonstrated for the convex case.

Overall, the reviewers find the contribution meaningful. The guidelines of design are of independent interest and further investigation.
 However, the presentation of the paper could be improved, and the paper would benefit from enhancing the numerical experiments, which the authors have done. Additionally, there are several works on adaptive step sizes based on the Barzilai–Borwein method that the authors need to discuss in the final version.

Hu, Jinhui, et al. "The Barzilai–Borwein Method for distributed optimization over unbalanced directed networks." Engineering Applications of Artificial Intelligence 99 (2021): 104151.

Zhang, Xuexue, Sanyang Liu, and Nannan Zhao. "An accelerated algorithm for distributed optimization with Barzilai-Borwein step sizes." Signal Processing 202 (2023): 108748.

Emiola, Iyanuoluwa, and Chinwendu Enyioha. "Q-linear Convergence of Distributed Optimization with Barzilai-Borwein Step Sizes." 2022 58th Annual Allerton Conference on Communication, Control, and Computing (Allerton). IEEE, 2022.